# High sulphur dioxide deposition velocities measured with the flux/gradient technique in a boreal forest in the Alberta oil sands region

Mark Gordon[1], Dane Blanchard[2], Timothy Jiang[1*], Paul A. Makar[3], Ralf M. Staebler[3], Julian Aherne[2], Cris Mihele[3], Xuanyi Zhang[1]

[1]Earth and Space Science, York University, Toronto, M3J 1P3, Canada
[2]Environmental and Life Science, Trent University, Peterborough, K9L 0G2, Canada
[3]Air Quality Research Department, Environment and Climate Change Canada, Toronto, M3H 5T4, Canada
[*]Now at School of Environmental Sciences, Guelph University, Guelph, N1G 2W1, Canada

*Correspondence to*: Mark Gordon (mgordon@yorku.ca)

**Abstract.** The emission of $SO_2$ from the Athabasca oil sands region (AOSR) has been shown to impact the surrounding forest area. Recent studies using aircraft-based measurements have demonstrated that deposition of $SO_2$ to the forest is at a rate many times higher than model estimates. Here we use the flux/gradient method to estimate $SO_2$ deposition rates at two tower sites in the boreal forest downwind of AOSR $SO_2$ emissions. We use both continuous and passive sampler measurements and compare both techniques. The measurements infer $SO_2$ deposition velocities ranging from 2.1–5.9 cm s$^{-1}$ (when corrections are applied). There are uncertainties associated with the passive sampler flux/gradient analysis, primarily due to an assumed Schmidt number, a required assumption of independent variables, and potential wind effects. We estimate the total uncertainty as ±2 cm s$^{-1}$. Accounting for these uncertainties, the range of measurements is approximately double the previous aircraft-based measurements (1.2–3.4 cm s$^{-1}$) and are more than 10 times higher than model estimates for the same measurement periods (0.1–0.6 cm s$^{-1}$), suggesting that $SO_2$ in the AOSR region has a much shorter lifetime in the atmosphere than is currently predicted by models.

## 1 Introduction

Emissions of sulphur dioxide ($SO_2$) from the Athabasca oil sands region (AOSR) in Alberta, Canada and its subsequent wet and dry deposition to the surrounding boreal forest ecosystems may lead to soil acidification (Aherne and Shaw, 2010). Several studies in the AOSR have shown that total sulphur deposition has the potential to cause soil and surface water acidification (Whitfield et al., 2010; Cathcart et al., 2016) and exceedance of critical loads of acidity (Makar et al., 2018). However, ecosystem impacts within the AOSR are ultimately dependant upon dry $SO_2$ deposition velocities given the low rainfall volumes (Clair and Percy, 2015). In concert, the lifetime of $SO_2$ in the atmosphere may affect downwind ambient air concentration and human exposure (Wright et al., 2018).

Recent studies using aircraft-based measurements (downwind of oil sands production) have demonstrated that dry $SO_2$ deposition velocities in the AOSR could be between 1.7 and 5.9 times higher than previous model estimates (Hayden et al., 2021). Hayden et al. determined total deposition fluxes between multiple 2-dimensional (vertical and crosswind) flux screens created using interpolated aircraft-based wind and concentration measurements. The aircraft is flown in crosswind transects at various heights to determine the total advective flux passing through a screen, and the deposition flux is determined as the

difference in advective flux between screens following a Lagrangian trajectory. For the three flights analyzed by Hayden et al. (2021), the dry deposition velocities were 1.2, 2.4, and 3.4 cm s$^{-1}$. In contrast, model parameterizations suggested deposition velocities of 0.72, 0.63, and 0.58 cm s$^{-1}$ for the same respective flight areas and periods.

Alternatively, dry deposition can be calculated directly at a measurement location using eddy covariance; however, this requires a fast-response instrument that can make measurements at 1 Hz or faster. This is not possible with current $SO_2$

instrumentation that measure at frequencies < 0.2 Hz, which necessitates using a flux/gradient approach (e.g., Wu et al., 2016). Using this method, a deposition velocity of 4.1 cm s$^{-1}$ was calculated over a 3-day period at a tower site in Fort McKay in the AOSR (Hayden et al., 2021). Although the tower site was within the small town of Fort McKay (population 750), it was surrounded by wooded and grassy areas and can be considered a residential/rural site.

Similarly, passive samplers can also be used to measure vertical gradients over long periods (c.f. Quant et al., 2021 and

references therein). This is especially useful for remote locations, since the samplers are relatively inexpensive, easy to deploy, have weekly to monthly exposure periods, and require no power. Quant et al. (2021) describe four vertical gradient passive sampler installations to measure gaseous mercury. They consider these results semi-quantitative and estimated an uncertainty factor of four. The bulk of this uncertainty (a factor of three) was due to the estimation of an appropriate average turbulent diffusion coefficient ($K$).

Bolinius et al. (2016) assessed the uncertainty of turbulent fluxes with long-term gradient profile measurements using the modified Bowen ratio to determine eddy diffusivity ($K$) from heat flux measurements. They tested this theory with highly variable and bi-directional fluxes of $CO_2$ and water vapour and found that the gradient method resulted in fluxes that differed by factors of three for $CO_2$ and 10 for $H_2O$.

This study used measurements of $SO_2$ gradients in a boreal forest to determine dry $SO_2$ deposition velocities downwind of oil

sands production facilities. Measurements were made with both long-term (2-3 week exposure period) passive samplers and continuous in-situ gas analyzers at two tower locations in the same forest. Continuous $SO_2$ measurements demonstrated that the site was subjected to relatively strong, intermittent plumes of $SO_2$, which we assumed were not re-emitted from the forest (hence eliminating uncertainty due to bi-directional fluxes). Here we determine average values of turbulent diffusion coefficients ($K$) based on momentum flux and stability measurements, and we assess the uncertainties due to the long-term

averaging of these variables using continuous $SO_2$ gradient measurements. Following this approach, we determine a range of $SO_2$ dry deposition velocities. This paper is a companion paper to Jiang et al. (2022) and Zhang et al. (2023) which respectively investigate aerosol and ozone deposition at this site.

## 2 Methods

### 2.1 Site Location and Instrumentation

This study incorporates measurements made at two towers in a mature jack pine (*Pinus banksiana*) forest. The location of the towers relative to surrounding mines and processing facilities is shown in Fig. 1a. The York Athabasca Jack-Pine (YAJP) tower was located at 57.1225 N 111.4264 W. The second meteorological tower was operated by the Wood Buffalo Environmental Association (WBEA), a non-profit environmental group, which monitors pollutants in the AOSR. WBEA identified this tower as "1004". It is approximately 540 m directly south of the YAJP tower. The forest extends for at least 10

km in all directions. Images of the YAJP and WBEA (1004) towers and surrounding forest are shown in Figs. 1b and 1c, respectively. The ground beneath the forest is covered in reindeer moss (*Cladonia spp.*). Undergrowth vegetation in the area is limited to sparsely distributed blueberry bushes. The soil is sandy and well drained. The forest canopy height is approximated as 19 m tall (with the tallest trees in the area ranging from 16 m to 21 m in height). The canopy leaf-area index (LAI) density profile measured in the vicinity of the YAJP tower is shown for comparison in Fig. 1d. The total LAI near the

YAJP site is 1.17. Details of the LAI measurement technique are given in our companion paper (Zhang et al., 2023).

The village of Fort McKay is approximately 15 km to the NW of the site and the town of Fort McMurray is 40 km south. The site is surrounded by oil sand production facilities. These include thermal in-situ extraction, such as Husky Sunrise and Suncor Firebag (approximately 23 km NW and 34 km WNW respectively); open pit mining, such as Shell Jackpine (10 km North), Syncrude Aurora (20 km NNW), and Shell Muskeg (15 km NW); as well as combined mining and upgrading

facilities, such as Suncor (13.5 km South), Syncrude (18 km SW), and CNRL (30 km NW). The upgrading facilities produce significant $SO_2$ plumes (c.f., Gordon et al., 2015) which are intermittently brought to the tower sites when the winds are from the South and SW directions (see Section 3.2). Additionally, the Hammerstone limestone aggregate quarry is located 10 km NW of the tower.

The 1004 tower measured wind speed and direction at heights of 2, 16, 21, and 29 m and recorded the data as 1-hour

averages. The YAJP tower measured high-frequency wind data with a 3D sonic anemometer (Type A, Applied Technology Inc.) mounted at a height of 29 m. Between September 2017 and August 2018, a second anemometer (Type V) was mounted within the canopy at height of 5.5 m. All flux, wind, and temperature measurements from the YAJP tower were calculated in 30-minute periods, unless stated otherwise.

Between 4-8 June 2018, $SO_2$ measurements were made at a height of 2 m at the YAJP tower with a 43i Thermo Scientific

analyzer (herein referred to as "43i"). A Tethered Balloon system made measurements of $SO_2$ using modified ozonesondes up to a height of 300 m between 13-15 July 2018. Between 7-26 August 2021, $SO_2$ measurements were made at heights of 2 m and 29 m using two 43i instruments sampling though lengths of ¼" Teflon tubing. For the 29-m height, the residence time of the tubing was measured as 14 s (a flow rate of 1.0 LPM). Between 20 July and 31 August 2021 an Envea AF22e gas analyzer (herein "AF22e") measured $SO_2$ at a height of 2 m. The AF22e was solar powered when generator power was not

available. It was therefore operated for a longer timespan than the 19 days of 43i measurements in 2021. All 43i

measurements were at a frequency of 0.2 Hz, while AF22e measurements were once per minute. Calibration of the 43i instruments determined a standard deviation of less than 0.07 ppb at 0 ppb and less than 0.43 ppb at 80 ppb. The AF22e was not laboratory-calibrated but was corrected against the co-deployed 43i (sampling from the same inlet) for measurement values ranging from 0 to 60 ppb ($R^2 = 0.97$).

To investigate the vertical structure of $SO_2$ plumes, a Tethered-Balloon system was used to lift two ozonesondes. Following the technique outlined in Yoon et al. (2022), one ozonesonde sampled through a filter tube coated with KMnO4 solution (which absorbs $SO_2$) so that the difference between the two ozonesonde measurements gives $SO_2$ mixing ratio. The procedure for ozonesonde preparation and calibration are outlined in Yoon et al. (2022).

Passive samplers were deployed at the YAJP and 1004 towers over eight separate exposure periods between October 2020

and October 2021. The first two deployments were at YAJP, the third was a co-deployment at YAJP and 1004 for the same period, and the remaining five were at 1004. The deployments ranged in duration from 2 to 3 weeks. Samplers were mounted at heights of 2, 4, 8.5, 13.5, 18, and 23 m on the YAJP tower and heights of 4, 8, 13, 17.5, and 22 m on the 1004 tower. The samplers mounted at a height of 2 m on the YAJP tower were only used for 2 of the 3 deployments.

The devices used in this work were badge-type passive samplers (Blanchard and Aherne, 2019; Islam et al., 2016;

Zbieranowski and Aherne, 2012) that housed a Whatman 40 filter paper pretreated with a KOH solution (Hallberg et al., 1984; Salem et al., 2009). Following field exposure, filters were extracted in 10 ml of 0.3% hydrogen peroxide solution and the resulting sulphate ($SO_4^{2-}$) concentrations were determined via ion chromatography. The mass of $SO_2$ collected ($Q$ [µg]) on the filter paper was calculated following Zbieranowski and Aherne (2012) as

$$Q = V M_R (S_f - S_b), \tag{1}$$

where $S_f$ [µg l$^{-1}$] is the measured $SO_4^{2-}$ concentration in the extraction solution, $S_b$ is the average field blank $SO_4^{2-}$ concentration (three per deployment), $V$ [l] is the extraction volume, and $M_R = 6.67 \times 10^{-4}$ is the molar conversion from $SO_4^{2-}$ to $SO_2$. For this study, the average value of $S_b$ was 0.01 mg l$^{-1}$ $SO_4^{2-}$, compared to an average $S_f$ of 1.38 mg l$^{-1}$ $SO_4^{2-}$, hence the uncertainty due to blank subtraction was assumed to be negligible. The atmospheric concentration of $SO_2$ was then calculated from Q as

$$C = \frac{Q R_t}{A t}, \tag{2}$$

where $C$ is the concentration of $SO_2$ (µg m$^{-3}$), $A$ is the sampler cross-sectional surface area (m$^2$), $t$ is the exposure time (s), and $R_t$ is the total badge-sampler resistance (s m$^{-1}$). A study specific $R_t$ value was estimated through the co-deployment of passive samplers at five WBEA monitoring stations equipped with continuous $SO_2$ monitors. Refer to Appendix A for further detail regarding $R_t$ determination and WBEA sampling site information.

Samplers were deployed in duplicate or triplicate on the underside of rain shelters. Data variability was assessed by calculating the coefficient of variation (CV) among replicate samplers. To reduce vibration and sway with winds, a rigid mounting system was developed to attach the samplers to a pulley rope (YAJP) or metal cable (1004), as shown in Figure 2.

At the YAJP site, the samplers were attached to a 1.5 m plastic tube that was fixed to the pulley rope at the top and bottom of the tube. Three of the five tubes were guyed to the ground with strings for additional stability. The lowest (2-m height) sampler at the YAJP site (used for 2 of the 3 deployments) was fixed to the tower base. At the 1004 site, the pulley system used a metal cable loop. Here the mounting system was fixed to one side of the loop, while the other side of the loop passed though forked stabilizers to eliminate vibration and sway.

The passive samplers co-deployed at the five WBEA continuous monitoring stations allowed the evaluation of sampler performance. Comparison with corresponding continuous measurement data enabled the calculation of passive sampler bias (%) while a Spearman Rank-Order Correlation test was applied to test the level of agreement ($\alpha < 0.05$) between samplers. At the YAJP tower, the AF22e was operational for the duration of the 4th sampler exposure period and the 43i was operational for the duration of the 3rd sampler co-deployment at both the YAJP and 1004 towers. These comparisons are discussed in Section 3.3.

## 2.2 Flux/Gradient Methodology

Deposition fluxes were calculated from the passive-sampler mixing ratio gradients following the gradient flux method from a procedure outlined in You et al. (2021). The relationship between the $SO_2$ deposition flux ($F$, positive downward) and the gradient ($dC/dz$) is

$$F = K_C \frac{dC}{dz},$$ (3)

where the trace gas diffusion coefficient ($K_C$) and vertical concentration gradient ($dC/dz$) are modeled as constant throughout the height of the canopy, although this assumes that the flux divergence is insignificant in the canopy (equivalent to assuming all deposition is to the surface and not to the canopy elements). This approach has been demonstrated to reproduce deposition velocities by Wu et al. (2016) using gradients at heights of 16.5 m and 33 m in a 22-m high mixed-deciduous canopy. This mixed-deciduous forest had an LAI of 4.6, compared to the LAI of 1.17 at our boreal forest site, suggesting that the denser foliage would have a greater effect on the in-canopy gradient at the mixed-deciduous site relative to our boreal site. The approach was also demonstrated by Meredith et al. (2014) using gradients measured at heights of 24 m and 28 m in a nearly 24-m high temperate forest with an LAI of approximately 4. Here we determine the concentration gradient using a least-squares fit to the measured 5-point profile within and above the canopy. Although some profiles had 6 points, the lowest measurement is not used in these cases for consistency in the analysis. We also compare this to a 2-point concentration gradient determined using the two highest measurement heights. As the LAI density distribution in Figure 1d and the supplementary Figure S1 demonstrate, the two upper measurement heights (18 m and 23 m at YAJP or 17.5 m and 22 m at 1004) can be considered above-canopy relative to the canopy height of 19 m. Results from both gradient calculation techniques are compared in Section 3.3.

While $K_C$ was not measured, the momentum diffusion constant ($K_{M,G}$) can be determined through the momentum flux/gradient relationship as

$$u_*^2 = K_{M,G} \frac{du}{dz},$$ (4)

where $u_*$ is the friction velocity (measured at a 29 m height) and the wind speed gradient ($du/dz'$) is approximated as $\Delta u/\Delta z$ from the wind velocity difference between heights of 29 m and 5.5 m. The uncertainty due to this approximation is investigated below. Since the wind speed gradient was only measured between September 2017 and August 2018, the 2020 – 2021 measurements require a parameterization of the diffusion constant. Ignoring the effects of the canopy on diffusion,

Prandtl's mixing length model is adjusted for stability to give (Garratt, 1994)

$$K_{M,P} = \frac{\kappa z_m u_*}{\phi},$$ (5)

where $\kappa = 0.4$ is the von-Karman constant, $z_m$ is the flux measurement height, and the stability parameter ($\phi$) can be determined from the Obukhov length ($L$) following Garratt (1994) as

$$\phi = \begin{cases} \left(1 - 16(z/L)\right)^{-1/4} & -5 < z/L < 0 \\ 1 + 5(z/L) & 0 < z/L < 1 \end{cases},$$ (6)

Between 23 September 2017 and 2 August 2018, two anemometers were functional on the YAJP tower and a gradient ($\Delta u/\Delta z$) was measured. For this period, we calculated $K_{M,G}$ from the flux/gradient method (Eq. 4) and compared this to the parameterization of $K_{M,P}$ from Eq 5 and 6. A least-squares fit to all the 30-min values of $K_{M,P}$ as a function of $K_{M,G}$ over the ~10-month period gave a slope of 2.6 with $R^2 = 0.83$ (Fig. 3), which supports the use of the $\Delta u/\Delta z$ approximation. Hence the diffusion can be more accurately parameterized as $K_{M,P} = \kappa z_m u_* / (2.6 \phi)$, which is simplified to $K_{M,P} = \kappa z'_m u_* / \phi$,

where $z'_m = z_m/2.6 = 11$ m. We note that the height of $z'_m = 11$ m lies between the two measurement heights (5.5 and 29 m), which supports the use of this technique. As demonstrated in Fig. 3, values of $K_{M,G} > 5$ m$^2$ s$^{-1}$ will be underestimated by this parameterization; however, less than 6% of the $K_{M,G}$ measurements in the 10-month period are greater than 5 m$^2$ s$^{-1}$. Other parameterizations (not shown here) that account for the effects of the canopy on mixing and turbulence are described in Stroud et al. (2005), Wu et al. (2016), and Makar et al. (2017). While these parameterizations provide greater accuracy for

vertically resolved modeling within the canopy, they do not improve the correlation between $K_{M,P}$ and $K_{M,G}$ for these data. This is likely because we are here approximating diffusion with a single value throughout the canopy.

The trace gas diffusion constant can be related to the momentum diffusion constant by the turbulent Schmidt number as

$$Sc = \frac{K_M}{K_C},$$ (7)

Schmidt numbers determined in previous studies demonstrate a range of values. Similarly, Flesch et al. (2002) measured

values between 0.17 and 1.34, while Gualtieri et al. (2017) report values in experimental and numerical studies between 0.1 and 1.3. You et al. (2021) defined a modified $Sc$ that incorporates the stability parameter ($z/L$) and found that the value varied between 0.04 and 2.90. The average values (or values determined by least-squares fitting) from Flesch et al. (2002),

You et al. (2021), and Gualtieri et al. (2017) are $Sc = 0.6$, 0.74, and 0.99 respectively. Here we use an average of 0.8 and estimate the uncertainty in the calculated values based on the 0.6 to 0.99 range of values.

With these assumptions, Equations 3 to 7 are combined to give the deposition flux as

$$F = \frac{\kappa \, z'_m}{Sc} \frac{u_*}{\phi} \frac{dC}{dz}. \tag{8}$$

## 2.3 Aerodynamic Resistance

The total resistance to pollutant deposition at height z ($r_{t,z}$) is modeled as the sum of the aerodynamic ($r_a$), quasi-laminar sublayer ($r_b$), and bulk surface ($r_c$) resistances. The deposition velocity is the inverse of the total resistance as

$$v_{d,z} = \frac{1}{r_{t,z}} = \frac{1}{r_a + r_b + r_c} = \frac{F}{C_z - C_0}, \tag{9}$$

where $C_z$ and $C_0$ are the concentrations at height z and at a compensation point, respectively. Typically, a zero concentration is assumed at the compensation point ($C_0 = 0$) either within the soil or the leaf, and the deposition velocity ($v_{d,z}$) can be related to the flux as $v_{d,z} = F/C_z$.

Our measurements in this study were limited to a height of 23 m and are therefore a calculation of $v_{d,23\mathrm{m}}$. We compared

these values to values determined by the GEM-MACH deposition parameterization (described in Section 2.5 below), which are also determined at a height of 23 m.

The deposition velocities in Hayden et al. (2021) were initially determined from aircraft measurements at heights > 150 m and then adjusted to a height of 40 m to give $v_{d,40\mathrm{m}}$. This accounts for reduced aerodynamic resistance between 40 and 150 m. Hayden et al. (2021) state that this extrapolation is considered their largest source of uncertainty. For comparison of the

deposition velocity calculated form YAJP and 1004 tower measurements, we added the aerodynamic resistance between heights of 23 m and 40 m, which can be calculated by integrating Eq. 3 between these two heights with $K_c = \kappa \, z \, u_* / Sc \, \phi$, and averaging to give

$$r_{a,23-40\mathrm{m}} = \frac{Sc}{\kappa} \langle \frac{\phi}{u_*} \rangle \ln\left(\frac{40}{23}\right), \tag{10}$$

This is added to the total resistance to give $v_{d,40\mathrm{m}} = \left(r_{t,23\mathrm{m}} + r_{a,23-40\mathrm{m}}\right)^{-1}$. The uncertainty associated with the added

resistance is discussed below.

## 2.4 Deposition Velocity Calculation

The total deposition can be calculated combining Equations 8, 9, and 10. As discussed in Section 2.2, the gradient ($dC/dz$) is determined using either a least-squares fit to 5 measurement heights (the 5-point gradient), or only using the two above-canopy measurement heights (the 2-point gradient). Using the 2-point gradient means that all uptake resistance ($r_{t,23\mathrm{m}}$) is

below the gradient. However, due to uncertainty in the measured value of $C$ using the passive samplers, there is higher

uncertainty associated with a 2-point gradient measurement. This uncertainty can be reduced by calculating the gradient as a least-squares fit to the five values of $C$ at all measurement heights. However, there are likely sinks in the region over which the 5-point gradient is estimated. As Figure 1d demonstrates, most of the leaf area is closer to the surface and the mean canopy height (50% total LAI) is 11.5 m. Hence, the deposition velocity is calculated with the 5-point gradient assuming that the error in the calculated gradient due to sinks throughout the canopy is small compared to the uncertainty in a 2-point gradient measurement. Both approaches are compared in Section 3.3.

The use of long-term passive samplers (2 to 3 weeks in duration) to determine the gradients necessitates time-averaging the equations. If it is assumed that $K_M$ (a function of $u_*/\phi$), the concentration ($C$), and the gradient ($dC/dz$) are all independent variables, this gives

$$v_{d,40\text{m}} = \left(\left(\frac{\kappa\, z'_m}{Sc}\langle\frac{u_*}{\phi}\rangle \frac{1}{\langle C\rangle}\langle\frac{dC}{dz}\rangle\right)^{-1} + \frac{Sc}{\kappa}\langle\frac{\phi}{u_*}\rangle \ln\left(\frac{40}{23}\right)\right)^{-1},\tag{11}$$

where the angle brackets $\langle\ \rangle$ indicate time-averaging over the sampling period. This assumes that there is no correlation between the stability-corrected friction velocity ($u_*/\phi$), the concentration ($C$), and the concentration gradient ($dC/dz$), since they are averaged separately in the equation. If $u_*/\phi$, $C$, and $dC/dz$ are correlated, the assumption of independent variables will introduce an error in this flux estimation (since $\langle u_*/\phi\, C\, dC/dz\rangle \neq \langle u_*/\phi\rangle\, 1/\langle C\rangle\, \langle dC/dz\rangle$). In order to estimate the error associated with the assumption of independent variables, we also calculate the deposition velocity (in Section 4.1) using a time series of 30-minute average, concurrent friction velocity, stability, and concentration measurements (using the high-frequency, $SO_2$ gradient measurements made with the two 43i instruments in August 2021), which does not require long-term averaging of these terms.

During some of the long-term averaging periods, due to a lack of sunlight to charge the batteries or instrumentation failure, a complete time-series of turbulence measurements ($u_*/\phi$) was not available. In these cases, missing values of $u_*$ were filled using the hourly-averaged wind speed ($U$) at the 1004 tower, which demonstrates a strong correlation ($R^2 > 0.8$) with $u_*$ at the YAJP tower. Missing heat flux values (to determine Obukhov length, $L$) were filled based on a median diurnal pattern determined from the available measurements. The extent of each case of turbulence data replacement is presented in the Results section below and the uncertainty based on the parameterization is discussed in Section 4.1.

## 2.5 GEM-MACH Deposition Parameterization

The GEM-MACH deposition parameterization used to compare to our measured values is described in Makar et al. (2018). The reader is referred to their Supplement S1 (their Equations S.1–S.20) for a detailed description. Very briefly, the parameterization accounts for aerodynamic ($r_a$), quasi-laminar ($r_b$), and bulk surface resistance ($r_c$), which includes resistances associated with soil, canopy, mesophyll, cuticle, and stomatal surfaces as well as resistance to buoyant convection. Here we model the forest as an evergreen needleleaf. The parameterization is a function of temperature, relative humidity, atmospheric stability, solar radiation, and $CO_2$ mixing ratio. For these values, we used measurements from the

YAJP and 1004 towers and we calculated deposition velocity (herein $v_{d,\text{GEM}}$) for the time periods coincident with the profile measurements.

As discussed above, the parameterized GEM-MACH deposition velocity values were significantly lower than the observations of Hayden et al (2021) for the same time periods and locations. Hayden et al. used a Monte Carlo analysis of the GEM-MACH deposition algorithm to demonstrate that the most likely cause of underestimation was in the standard model assumption that concentration of hydrogen ions on the mesophyll, cuticle and exposed surfaces corresponded to a neutral pH (6.68). The Oil Sands facilities are known sources of significant base cation emissions (the neutralizing impact of the base cations on acidifying deposition was noted in Makar et al (2018)). Hayden et al (2021) showed that the increase in surface pH associated with deposited base cations could account for the discrepancy between modelled and measured $SO_2$ deposition velocities and fluxes. That is, $SO_2$ deposition close to the sources is likely being enhanced by the co-deposition of base cations.

## 3 Results

### 3.1 Passive Sampler Evaluation

An analyte detection limit was calculated as the product of standard deviation (five blanks per sampler batch) and the t-value for a 99.0% confidence critical value. The sampler batch detection limits (assigning shortest exposure length for conservative values) ranged between 0.016 to 0.022 ppb $SO_2$, with a study avg. of 0.019 ppb $SO_2$. A significant linear correlation ($R^2 = 0.95$; $\alpha < 0.05$) was observed between the 14 co-located passive samplers and WBEA continuous $SO_2$ measurements. The estimated sampler bias was low (2.3%), while replicate sampler variability remained low throughout the study period with a CV = 4.5%.

### 3.2 Characterizing the SO₂ Plume

A time series of $SO_2$ measurements is shown in Figure 4. These measurements demonstrate the intermittency of the plumes and range of values. There are also 10 days of data from 2018 not shown here which demonstrate a similar intermittency. Figure 5 plots all the measurements (including the 2018 measurements) with wind direction. This distribution demonstrates that most of the SO2 is transported from the Suncor (13.5 km at 195º) and Syncrude (18 km at 225º) stacks.  All the measurements between 1 and 10 ppb (green dots) outside the 160 º to 250º range are from one day in July with low winds and variable wind direction measured with the AF22e. This is maybe a recirculation event drawing back a Syncrude or Suncor plume from different directions.

Figures 4 and 5 demonstrate how winds bring elevated $SO_2$ levels from the south to south-west. Plumes of $SO_2$ with 5 ppb or higher appear on the majority of days, typically occurring between 09:00 and 18:00, although the duration of the plume exposure through the day can last from 1 to 8 hours. The skewness of the data (i.e., a majority of near-zero values with some

strong intermittent pulses of high $SO_2$ levels) have implications for how an average passive sampler value should be interpreted. For example, when comparing passive-sampler measurements at two locations, slight differences in wind patterns between the two locations could lead to vastly different average $SO_2$ values.

$SO_2$ measurements were made from a modified ozonesonde on a tethered balloon. These measurements are compared here to ground $SO_2$ measurements (43i) in Figure 6. The figure demonstrates decoupling and coupling of the different vertical layers of air in the boundary-layer above the forest. During the first half of the ascent (elevation shown by black line), the balloon and ground $SO_2$ measurements are nearly equal (red and green lines respectively), demonstrating a well-mixed boundary-layer. At the end of the ascent (~13:10), the sonde samples an $SO_2$ plume while the ground 43i samples clean air. The plume

begins to mix to the ground level near 13:25 and things become well-mixed near 13:40 with nearly equal ground and elevated $SO_2$ values. This demonstrates that plumes are not only intermittent due to horizontal variation in wind direction but can also vary considerably in the vertical direction. Our analysis of deposition using a flux/gradient approach assumes that when these horizontal and vertical variations are averaged over a 2 to 3-week period, a smooth vertical gradient is observed.

### 3.3 Deposition Velocity

The 9 passive sampler profiles (from 8 time periods) are listed in Table 1. Durations of the sample periods range from just over 12 days to more than 3 weeks. The sampler vertical profiles of $SO_2$ mixing ratio with height are shown in Fig. 7. The gradients ($dC/dz$) are determined either as a 2-point gradient above the canopy (to give $v_{d,23m}^+$) or from a least-squares fit to the 5-point gradient (to give $v_{d,23m}$). The mixing ratio ($C$) is determined from the highest sampler location. Using the 2-point, above-canopy gradients, the deposition velocities calculated with Eq. 8 range from 1.4 to 28.0 cm s$^{-1}$, with an average

of 9.5 cm s$^{-1}$. Using the 5-point gradients, the deposition velocities calculated with Eq. 8 range from 2.9 to 9.4 cm s$^{-1}$, with an average of 6.9 cm s$^{-1}$. The $R^2$ values for the least-squares fits are given in Table 1. Although the above-canopy, 2-point gradient results in a higher average deposition velocity, there is much higher variation across the 9 profile measurements, and the average values from each method (9.5 cm s$^{-1}$ and 6.9 cm s$^{-1}$) are not statistically different at a 55% confidence level (i.e. $\overline{v_d} \pm 0.75\ \sigma/\sqrt{n}$). Hence, we conservatively focus on the lower deposition velocities calculated with the 5-point

gradient determined by least-squares fit ($v_{d,23m}$), but note the high uncertainty associated with these measurements.

Adjusting the deposition velocities (calculated with the 5-point gradients) to a height of 40 m (from 23 m) reduces the range to 2.7 to 7.7 cm s$^{-1}$ with an average value of 5.9 cm s$^{-1}$. The difference between $v_{d,23m}$ and $v_{d,40m}$ for each profile ranges from 7 to 18%. The values measured here are approximately double those of Hayden et al. (2021) which range from 1.2 to 3.4 cm s$^{-1}$. As was seen in Hayden et al., the values are considerably higher than the GEM-MACH parameterization (Makar

et al. 2018) and an inference model used Hsu et al. (2016) for the AOSR, which range from 0.2 to 0.3 cm s$^{-1}$.

The AF22e and 43i analyzer measurements are compared to the passive sampler measurements in Figure 7 for coincident periods during Profiles 3a and 3b (two sampler installations at the same time at the YAJP and 1004 towers) and Profile 4 (1004 only). In both cases, measurements near the surface show good agreement with passive samples measurements

(accounting for the gradient of mixing ratio with height). During the Profile 4 period, the 43i measurements at a height of 29 m are much lower than the highest passive sampler measurement over the same period. Using the linear fit to the passive sampler gradient and assuming a linear interpolation of the gradient below and above the canopy, the mixing ratio at canopy height (19 m) would be approximately 1.37 ppb from the passive sampler profile versus 0.93 ppb from the 43i profile (a 32% difference). One potential reason for this discrepancy could be a dependence of the passive sampler resistant ($R_t$) on wind speed, which could lead to overestimation of passive sampler measured concentration above the canopy (where the wind speed is greater). This potential effect is investigated in greater detail in Section 4.2. The deposition velocity calculated with the 43i profile measurements (adjusted for aerodynamic resistance between 29 m and 40 m) is 4.4 cm s$^{-1}$, which is 68% of the 6.5 cm s$^{-1}$ deposition velocity determined by the passive sampler gradient over the same period (Profile 4, Table 1). This value is in close agreement with the flux/gradient measurement (4.1 cm s$^{-1}$) of Hayden et al. (2021).

## 4 Discussion

### 4.1 Uncertainty Analysis

There is approximately a factor of 2 difference between the range of deposition velocities reported here ($v_{d,40m}$ between 2.7 and 7.7 cm s$^{-1}$) and the aircraft-based measurements in the region (1.2 – 3.4 cm s$^{-1}$, Hayden et al., 2021), but in both cases the measured values are considerably higher than the GEM-MACH parameterized values and the values determined by an inference model for the AOSR region of Hsu et al. (2016). For comparison, the range of deposition velocities for different methods and studies are listed in Table 2. We note that the aircraft measurements of the Hayden et al. study covers a range of different forests and land types, including lakes, wetlands, and surfaces modified by oil sands extraction, waste, and tailings. The estimate made using a flux/gradient approach in Hayden et al. is from a 3-day period (4.1 cm s$^{-1}$) at a tower in the town of Fort McKay. This value is closer to our measured values, but still 30% less than our average value. Below we investigate whether measurement uncertainty based on assumptions in the methodology might be responsible for the observed differences.

As discussed above, the use of the 5-point gradient effectively moves the total resistance (including aerodynamic, quasi-laminar sublayer, and bulk surface resistances) to the ground level, following the "big-leaf" assumption typically used by regional and global-scale models, as opposed to a vertical distribution of uptake throughout the canopy. While there is uncertainty associated with this assumption which is difficult to quantify, the average deposition velocity calculated with the 2-point, above-canopy gradient is greater than the average deposition velocity calculated with the 5-point gradient, suggesting that the deposition velocity calculated with the 5-point gradient may be a conservative estimate (although the averages are within $\pm 0.75$ standard errors and hence not significantly different).

The deposition velocities ($v_{d,40m}$) from the 9 profiles can be compared to a normal distribution, with 78% (7 values) within one standard deviation ($\sigma$) of the mean, 1 value (Profile 8) 1.3$\sigma$ higher than the mean and 1 value (Profile 5) 2.4$\sigma$ lower than

the mean. The anomalous deposition velocity of 2.9 cm s$^{-1}$ for Profile 5 is due to a combination of a weak gradient and high mixing ratio relative to the other profiles (Fig. 7). The fit to the profile is moderate ($R^2 = 0.72$), but not the weakest fit. Meteorological conditions shown in Appendix A (Fig. A1) demonstrate some rainfall (> 25 mm) and some cloudy, humid conditions, but similar conditions are seen in other profile periods (e.g. rain in profile periods 1 and 3, and clouds and high humidity in profile periods 6 and 8). Hence the reason for this anomalous value is unknown.

The two highest deposition velocities ($v_{d,40m} = 6.9$ and 7.7 cm s$^{-1}$ for Profiles 1 and 8) are from periods when most of the turbulence data were unavailable (61.5% and 96% missing respectively). As discussed in Section 2.2, $u_*$ and the heat flux were parameterized based on measured wind speed and an assumed diurnal profile. Hence there will be greater uncertainty in these measurements. As an example, recalculating all the deposition velocities with completely parameterized $u_*$ and heat flux results in a range of $v_{d,40m}$ from 3.2 to 7.8 cm s$^{-1}$ with an average of 6.3 cm s$^{-1}$. This suggests that the parameterization

of missing turbulence data may lead to an approximate 10% overestimation of $v_d$.

Eq. 5 was used to parameterize $K_M$ due to a lack of wind gradient measurements during the passive sampler measurement period. As demonstrated by Fig. 3, this introduces some uncertainty to the measurements. Previous measurements at the site demonstrate deviation from a linear fit for values of $K_{M,G} > 5$ m$^2$ s$^{-1}$, which corresponds to values of $K_{M,P} > 12.5$ m$^2$ s$^{-1}$ or $u_*/\phi_M > 2.8$ m s$^{-1}$. By comparison, the maximum value reached during Profile 3b (for example), is $u_*/\phi_M = 1.4$ m s$^{-1}$ (or

$K_{M,P} = 6.2$ m$^2$ s$^{-1}$), so the effect of this deviation from the linear fit should not be significant. The uncertainty can be approximated from the standard error of the slope, which is 0.017. From Eq. 8, this standard error gives an uncertainty of less than 2% in the final $v_d$ estimate using a 95% confidence interval.

Since $v_d$ is inversely proportional to the Schmidt number, our assumption of a constant value of $Sc = 0.8$ can be assessed against other published average values of $Sc = 0.6$ and 0.99. Using a value of $Sc = 0.6$ would result in an increase of 33% in

$v_d$, while a value of $Sc = 0.99$ would result in an 20% decrease. You et al. (2021) suggest a Schmidt number which is a function of stability (based on the ratio of measured momentum diffusivity coefficient to measured concentration diffusivity coefficient). We use the turbulence data from Profile 3b to investigate the use of this variable Schmidt number. Profile 3b is chosen because of the strong fit of the slope ($R^2 = 0.91$) and the completeness of the turbulence data over that period (Table 1). For this reanalysis, $Sc = 0.08 + 3.13 \times 10^{-9}$ exp (($z/L$ +19.5)/1.008) for $z/L <$ -0.18 and $Sc = 0.74$ for $z/L \geq$ -0.18

(following You et al.2021). It is noted we are using a parameterized momentum diffusivity coefficient ($K_{M,P}$) to estimate the gradient momentum diffusivity coefficient ($K_{M,G}$) and the parameterization (Eq. 5) includes a correction for stability. Hence, this method includes two stability corrections: one for the assumption of $K_{M,P} = K_{M,G}$ and one for the variability observed in $Sc = K_{M,G}/K_C$. Regardless of this complication, the use of this variable Schmidt number results in a decrease in $v_{d,40m}$ of 50% (to 2.7 cm s$^{-1}$) for Profile 3b.

As discussed in Section 2.4, the assumption that $u_*/\phi, C$, and $dC/dz$ are independent variables will result in an error if they are correlated. We investigate this assumption by calculating the deposition velocity using a continuous time series of 30-min averages of the friction velocity, stability, and SO$_2$ observations (measured with the 43i instruments at 2 m and 29 m

heights). The use of 30-min averages is long enough to give confidence in the calculated turbulent statistics but should allow for co-variation in $u_*/\phi$, $C$, and $dC/dz$. This allows for a calculation of $v_{d,40\text{m}}$ for each 30-min period. The average of these

30-min $v_{d,40\text{m}}$ values can then be compared to an average over the entire period following Eq. 11 (i.e. $\langle \frac{\kappa\, z_m}{Sc} \frac{u_*}{\phi} \frac{1}{C} \frac{dC}{dz} \rangle$)

compared to $\frac{\kappa\, z_m}{Sc} \langle \frac{u_*}{\phi} \rangle \frac{1}{\langle C \rangle} \langle \frac{dC}{dz} \rangle$.)  In order to avoid large numbers caused when $C \approx 0$, the analysis is restricted to times when a plume is present using a criteria of $C > 1$ ppb $SO_2$ (cf. Fig 4). This gives 73 30-min averages for analysis when $SO_2$ mixing ratio and turbulence data are available (no filling of missing turbulence data is applied here). The average of all the 73 30-min deposition velocities is $v_{d,40\text{m}} = 3.3$ cm s$^{-1}$.  The deposition velocity calculated using the averages of $\langle u_*/\phi \rangle$, $\langle C \rangle$, and

$\langle dC/dz \rangle$ for the same 73 30-min values is $v_{d,40\text{m}} = 4.4$ cm s$^{-1}$. This indicates that the assumption of independent variables required by long-term averaging could lead to a 30% overestimation of the deposition velocity. Here we define a corrected deposition velocity (which assumes the 30% overestimation applies to all measurements) as $v_{d,40\text{m}}^* = v_{d,40\text{m}}$ / 1.3. Applying this correction to the range of deposition velocities listed in a Table 1 gives a corrected range of 2.1 to 5.9 cm s$^{-1}$, with an average of 4.6 cm s$^{-1}$.

**4.2 Wind Speed Effects**

Some of the profiles shown in Fig. 7 have mixing ratios near or above the canopy which are much higher than the within-canopy values. Although the within-canopy values demonstrate an increase with height, for many profiles that increase is much more pronounced across the canopy top. For example, Profiles 1 and 4 show a sharp increase in mixing ratio above the canopy, while Profiles 3a and 8 show a sharp increase in the two highest measurement heights (relative to measurements in

the sub-canopy). These 4 profiles (1, 3a, 4, and 8) demonstrate higher than average deposition velocities ($\geq 6.3$ cm s$^{-1}$). By comparison, Profiles 2 and 3b show the best agreement to the linear least-squares fit ($R^2 = 0.99$ and 0.91 respectively) and have the lowest deposition velocities (5.3 and 5.4 cm s$^{-1}$). These results suggest that the greater-than-average increase in mixing ratio at the top of the canopy is associated with higher estimates of $v_d$ and a lower correlation of the profile with a linear fit.

Wind speed can affect the sampling rate of badge type samplers; however, the effect is reduced by the diffusion membrane (Plaisance, 2011) and use of a wind shield (Masey et al., 2017). Hofschreuder et al. (1999) noted that with proper sampler and draught shield design the influence of wind speed can be reduced to less than 10%. Although our sampler concentrations were corrected using samplers mounted at WBEA stations coincident with continuous gas analyzers, the wind conditions at these stations might show significant differences compared to the canopy, and the increase in wind speed near the canopy top

may have significant effect on the measured $SO_2$ gradient.

Using the wind speeds profiles measured at 1004, the correlation of the above-canopy concentration gradient ($dC/dz$) between 18 and 23 m can be compared to wind data for each profile period. The values of $dC/dz$ for each period show no correlation ($0.001 < R^2 < 0.03$) with either: the average wind speed gradient ($dU/dz$ between 16 and 29 m); the average

wind speed at 29 m; or the variance in hourly wind speeds at 29 m (all measured over the same time periods as the profiles).

If wind speed had a direct effect on the sampler uptake, leading to higher measured $SO_2$ mixing ratios for higher wind speeds, then a stronger correlation would be expected between the upper concentration gradient across the canopy top, where the wind gradient is largest. However, the low correlation between $dC/dz$ and various variables related to wind speed suggests that the vertical gradient of wind speed between the sub-canopy and above the canopy does not have a significant effect on the measured $SO_2$ mixing ratios.

In order to assess passive sampler performance, passive samplers were deployed at 5 WBEA continuous monitoring sites over five 2-week periods, resulting in 14 comparisons between passive sampler and continuous measurements (the number of deployment sites for the five periods were 2, 3, 4, 4, and 1 respectively). During these periods, wind speeds at the site (measured at a 10-m height) ranged from an average of 1.3 m s$^{-1}$ to 4.0 m s$^{-1}$. The passive sampler measurement error (the difference between the passive sampler and continuous measurements) ranged from −0.36 to 0.29 ppb. If the passive sampler

resistance ($R_t$) varies with wind speed it would be expected that this error would correlate with the average wind speed for each sampler comparison. However, the correlation between sampler error and average wind speed is $R^2 = 0.003$. Similarly, the correlation between sampler error and either maximum wind speed or wind speed variance is $R^2 < 0.003$. Hence, these results do not suggest a strong influence of wind speed on the measured passive sampler concentration.

**5 Conclusions**

The continuous $SO_2$ gradient measurements made by gas analyzers above and below the canopy were averaged to a frequency of 30 min to assess the assumption of independent variables that is required for the averaging of turbulence measurements to determine deposition rates from long-term passive sampler measurements. For the 19-day period analyzed, these results suggest an overestimation of $v_d$ of 30% due to the assumption of independent variables. Assuming this overestimation is the same for all profile periods, the corrected range of deposition velocities would be 2.1 to 5.9 cm s$^{-1}$ with

an average of 4.6 cm s$^{-1}$.

There is disagreement between the passive sampler gradient and the continuous $SO_2$ 43i gradient measurements made by gas analyzers over the same period. The predicted mixing ratio at canopy height from the passive sampler profile is 47% higher than the predicted mixing ratio at canopy height from the continuous 43i gradient profile (assuming a linear profile in each case). This could be partially due to wind effects causing overestimation of the passive sampler concentrations. However,

our investigation of wind speed effects on the measurements shows no correlations between measured wind speed and concentration gradients or the error in $R_t$. The difference in slopes ($dC/dz$) between the continuous measurements and the passive sampler profiles results in a nearly 50% difference in predicted deposition velocity. Correcting the passive sampler deposition estimation for the 30% overestimation due to the independent variable assumption gives $v_d = 5.0$ cm s$^{-1}$. Hence, given the uncertainties involved, these two measurement methods give comparable deposition velocity estimates for the

same time period within ±1 cm s$^{-1}$.

The predicted range of deposition velocities (2.1 to 5.9 cm s$^{-1}$ accounting for independent variables) is higher than the range of values of 1.2 to 3.4 cm s$^{-1}$ determined by Hayden et al. (2021) using aircraft measurements, although these values are close to a deposition velocity of 4.1 cm s$^{-1}$ reported in the Hayden et al. (2021) study determined using a flux/gradient approach from a tower located in Fort McKay. However, these results support the conclusion of Hayden et al. (2021) that

deposition to forest surfaces is likely underestimated in regional and global chemical transport models as both sets of results are considerably higher (by an order of magnitude in our case) than parameterized values.

A near-unity value of $Sc = 0.99$ would results in a 20% reduction in the estimated values and the use of a variable $Sc$ parameterization based on stability results in an 50% decrease in the estimated deposition rate. Hence, there is substantial uncertainty ($\pm1.5$ cm s$^{-1}$) based on the assumed Sc value.

The use of passive sampler gradients to determine fluxes is relatively new and previous studies (e.g., Quant et al., 2021) have suggested large uncertainties. The uptake to the passive samplers may depend on properties such as wind speed and temperature, which also have strong vertical gradients, especially within and above a canopy. More study of known fluxes is required to fully quantify the uncertainties of this measurement technique.

The flux/gradient method does not account for flux divergence through the canopy and is equivalent to assuming that the

total deposition occurs near the ground level only. Although the uncertainty associated with that assumption is difficult to quantify, calculating the gradient using above-canopy measurements results in a higher average deposition velocity and a greater disparity between our measurements and those determined by Hayden et al. (2021). However, there is high variability between the deposition velocities calculated from the different profiles with the above-canopy measurements, suggesting high uncertainty in the average measured value. We note that the deposition velocities calculated with the 5-point

measurement (throughout and above the canopy) show good agreement with the Hayden et al. (2021) tower measurements located in a relatively clear and residential/rural area in Fort McKay, suggesting that the error associated with the vertical positioning of the uptake elements may be small. To quantify this uncertainty, the use of a high-resolution, one-dimensional canopy model (such as the model used in the Zhang et al., 2023) which can correctly model the vertical distribution of uptake is recommended for future studies.

Despite the uncertainties in the measurements, all the measurements for the AOSR in this study and the Hayden et al. (2021) study are significantly greater than model parameterizations. These results suggest much shorter lifetime of $SO_2$ in the atmosphere and significantly more sulphur deposition to the type of environment studied here than has previously been modeled, in agreement with the conclusions of Hayden et al. (2021). These results support the hypothesis discussed in Hayden et al. that $SO_2$ co-deposition with base cations may influence local $SO_2$ deposition fluxes. This has consequences for

both the contribution of sulphur to atmospheric aerosols (which affect climate forcing) as well as ecosystem health of the boreal forest environment. The discrepancy between these measured deposition velocities and parameterizations for this region suggests that further study is required to investigate these differences.

## Appendix A

Passive samplers were co-deployed at five WBEA active monitoring stations in order to estimate the $R_t$ value for the filter solutions during each exposure (Zbieranowski and Aherne, 2012)

$$R_t = \frac{A_c \, A \, t}{Q},$$ (A1)

Where $A_C$ is the active sampler measured $SO_2$ concentration (µg m$^{-3}$) during the exposure period, and the remaining variables are described in section 2.1. Passive sampler concentrations were calibrated using the average $R_t$ observed over the entire study period.

WBEA sampling site information, listing station names, locations, and links to website description (last accessed Sept, 2022) are listed in Table A1.

Meteorological data is shown in Fig. A1 for the 8 sampling periods on the YAJP and 1004 towers (as listed in Table 1). These data demonstrate the variation in precipitation, wind speed, sunlight (as demonstrated by photosynthetically active radiation), relative humidity and temperature. 3 of the 8 sampling periods showed significant rainfall (1, 3, and 5). Sampling period 2 showed the coldest temperatures (reaching −20°C), while sampling period 3 was the warmest (reaching 30°C).

### Data Availability

The data described in this study are available at https://doi.org/10.5683/SP3/OQFZXZ (Gordon, 2023).

### Author Contributions

MG wrote the original draft of this work and performed the analysis. DB prepared and analyzed the passive samplers. MG, DB, TJ, and XZ performed the field studies. PM acquired funding. CM provided instrument support and calibration. All authors provided input, reviewed, and edited the work.

### Competing Interests

The contact author has declared that none of the authors has any competing interests.

### Acknowledgements

We acknowledge the shared data, technical support, and assistance of the Wood Buffalo Environment Association (WBEA) of Alberta. We thank James Flynn at the University of Houston (Texas) and Mark Spychala at St. Edward's University

(Texas) for joining our field study in 2018 to test their novel SO$_2$ sonde (see Yoon et al., 2022) and for sharing the sonde data measured from our tethered balloon.

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

Table 1: Details of the passive sampler installations and the resulting deposition velocity estimates, calculated using the concentration gradient either above the canopy ($v^+_{d,23m}$) or throughout the canopy ($v_{d,23m}$). The available turbulence data indicates the completeness of the $u_*$ and heat flux data in that period. Deposition velocities calculated from the GEM-MACH parameterization ($v_{d,GEM}$) over the same periods are also shown (determined at a height of 23 m). The average value (and standard deviation) of $v_d$ from all the profiles is shown in the bottom row. Deposition velocities adjusted to a height of 40 m ($v_{d,40m}$) are shown for comparison with Hayden et al. (2021).

| Profile ID | Instal. Date | Duration (days) | Location | Turb. data | Profile $R^2$ | [$SO_2$] (ppb) | $v^+_{d,23m}$ (cm s$^{-1}$) | $v_{d,23m}$ (cm s$^{-1}$) | $v_{d,GEM}$ (cm s$^{-1}$) | $v_{d,40m}$ (cm s$^{-1}$) |
|---|---|---|---|---|---|---|---|---|---|---|
| 1 | 2020-10-07 | 13.9 | YAJP | 28.5% | 0.73 | 0.37 | 28.0 | 8.0 | 0.14 | 6.9 |
| 2 | 2021-03-09 | 14.9 | YAJP | 89.2% | 0.99 | 2.24 | 4.0 | 6.0 | 0.21 | 5.3 |
| 3a | 2021-07-20 | 14.0 | YAJP | 100% | 0.85 | 1.55 | 12.8 | 7.5 | 0.50 | 6.3 |
| 3b | 2021-07-20 | 14.0 | 1004 | 100% | 0.91 | 1.68 | 3.5 | 6.1 | 0.50 | 5.4 |
| 4 | 2021-08-03 | 20.9 | 1004 | 91.4% | 0.78 | 1.19 | 18.7 | 7.6 | 0.53 | 6.5 |
| 5 | 2021-08-24 | 21.1 | 1004 | 98.3% | 0.72 | 1.89 | 1.4 | 2.9 | 0.55 | 2.7 |
| 6 | 2021-09-14 | 14.9 | 1004 | 93.8% | 0.85 | 0.57 | 2.6 | 7.6 | 0.47 | 6.3 |
| 7 | 2021-10-01 | 13.0 | 1004 | 100% | 0.62 | 1.88 | 1.9 | 7.4 | 0.48 | 6.2 |
| 8 | 2021-10-14 | 12.2 | 1004 | 4% | 0.87 | 1.48 | 12.8 | 9.4 | 0.45 | 7.7 |
| Avg | | | | | | | 9.5 | 6.9 | 0.42 | 5.9 |
| (Std) | | | | | | | (8.7) | (1.7) | (0.14) | (1.3) |

Table 2: A comparison of deposition velocities measured in the AOSR with different measurement methods or parameterizations. The height refers to either measurement height (23 or 32 m) or an adjusted height of 40 m (which modifies the aerodynamic resistance term, $r_a$). Passive Gradient and Continuous Gradient refer to the flux/gradient method using either long-term passive samplers or continuous measurements, respectively. The Passive Gradient (corrected) method includes a correction based on a demonstrated overestimation due to the assumption of independent variables. The Continuous Gradient does not require this correction. The Aircraft method used mass-balance of $SO_2$ plumes at multiple locations downwind of the emissions source.

| Method | Variable | Height | Range (cm s$^{-1}$) | Source |
|---|---|---|---|---|
| Passive Gradient | $v_{d,23m}$ | 23 m | 2.9 - 9.4 | This study |
| Model Parameterization | $v_{d,23m}$ | 23 m | 0.1 - 0.6 | GEM-MACH, from Makar et al., 2018 |
| Model Parameterization | $v_d$ | † | 0.2 - 0.3 | NOAA-MLM, from Hsu et al., 2016 |
| Passive Gradient | $v_{d,40m}$ | 40 m | 2.7 - 7.7 | This study |
| Passive Gradient (corrected) | $v^*_{d,40m}$ | 40 m | 2.1 - 5.9 | This study |
| Continuous Gradient | $v_{d,40m}$ | 40 m | 3.3 | This study |
| Continuous Gradient | $v_{d,32m}$ | 32 m | 4.1 | Hayden et al., 2021 |
| Aircraft | $v_{d,40m}$ | 40 m | 1.2 - 3.4 | Hayden et al., 2021 |

† Given only as a "shallow sub-layer within the atmospheric constant flux layer"

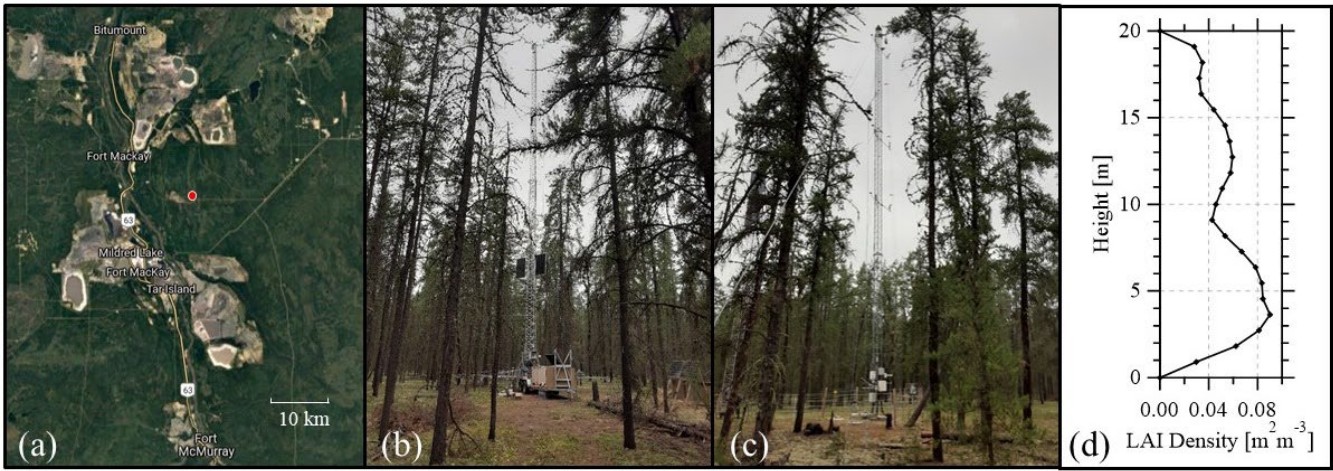

**Figure 1: (a) The study area and surrounding location showing the location of the towers (red dot) relative to oil sands mining and production facilities. (b) A photo of YAJP tower. (c) A photo of WBEA 1004 tower. (d) The LAI density profile near the YAJP station. Map image is © Google Maps. Photos taken by authors.**

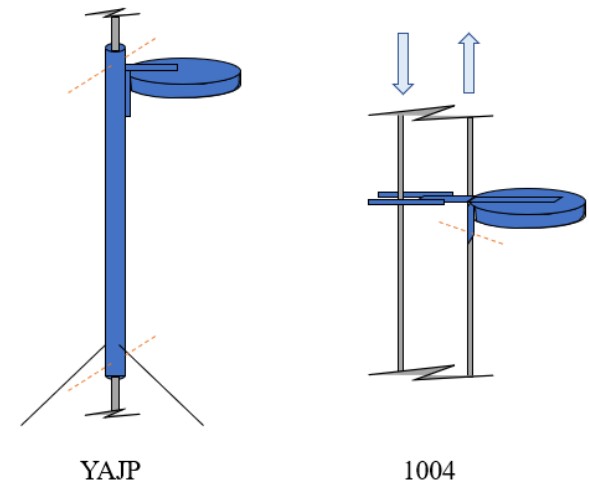


**Figure 2: Schematic of passive sampler mounts (not to scale). Grey lines indicate pulley rope (YAJP) or pulley cable loop (1004). Orange dashed lines show where the system is fixed against the rope or cable. The YAJP system used guy ropes and the 1004 system used a tong or forked support against the looped pulley cable to inhibit rotation. Multiple passive samplers were fixed to the underside of the rain shelter lids.**


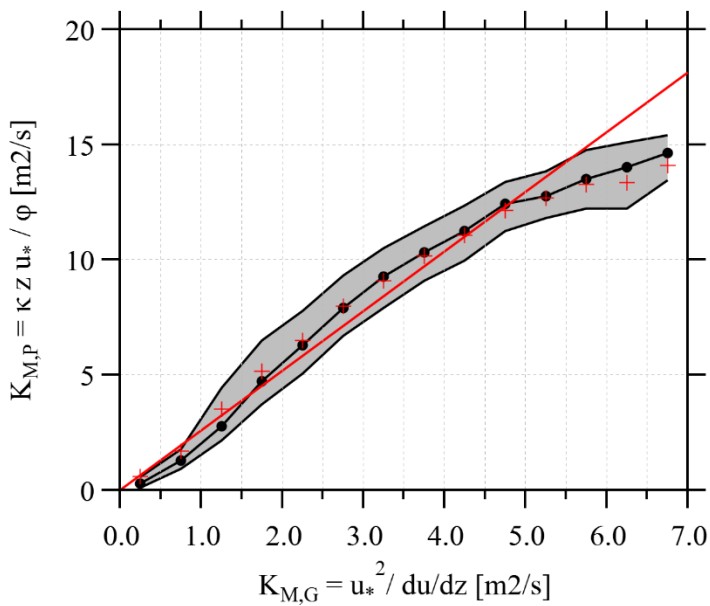

**Figure 3: A comparison of momentum diffusion coefficients ($K_M$) determined through flux/gradient method ($K_{M,G}$), compared to the parameterization of Eq. 5 and 6 ($K_{M,P}$). The parameterized values are binned by flux/gradient values. Black circles show medians, grey shading shows 25th and 75th percentiles, red pluses show averages, and the red straight line shows a least-squares fit to all 30-min data.**


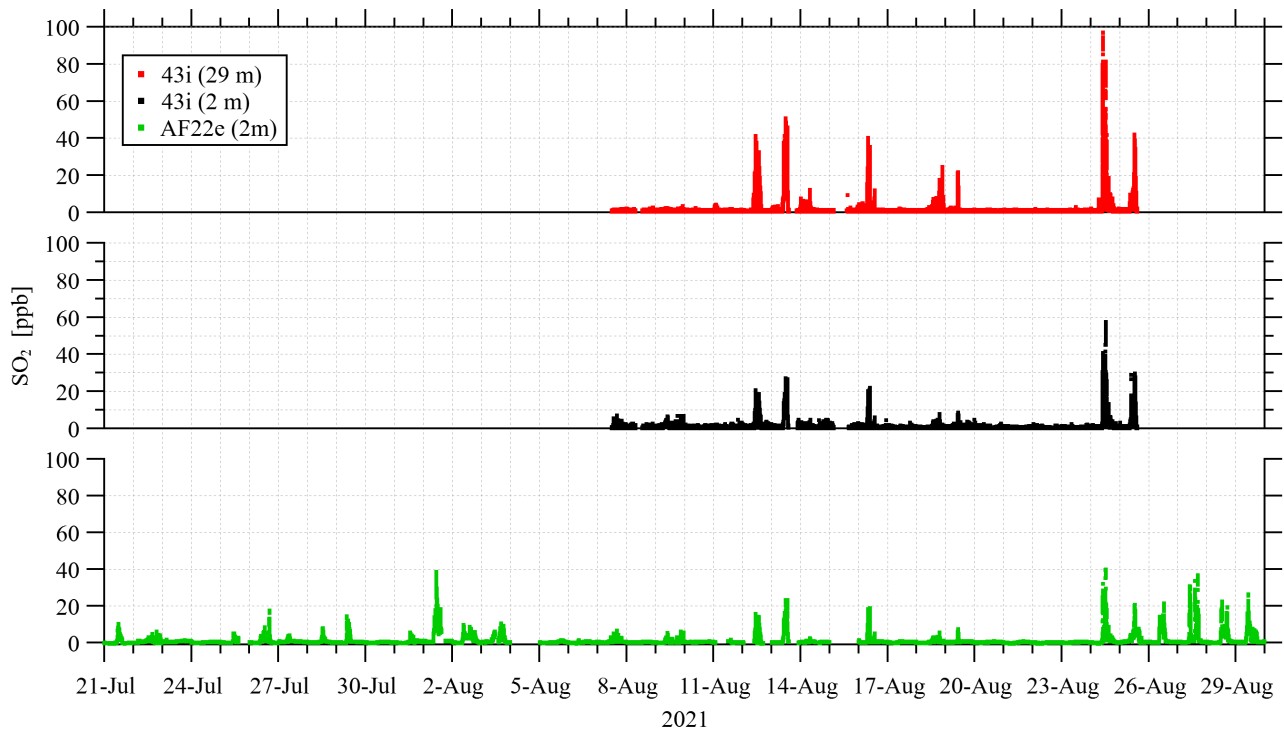

**Figure 4: SO₂ measurements from gas analyzers at YAJP tower. 43i measurements are every 5-sec, AF22e are every minute.**


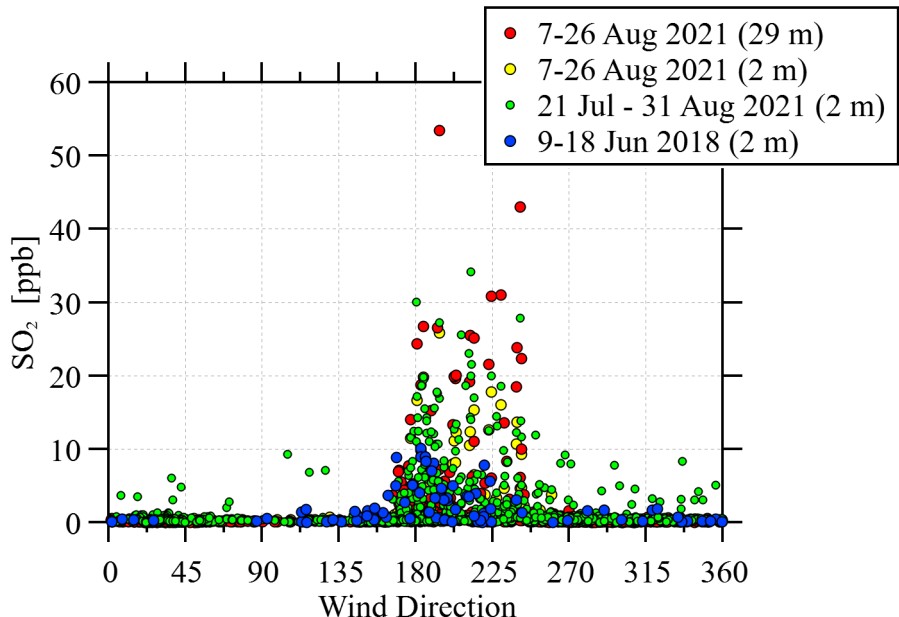

**Figure 5: SO₂ measurements as a function of wind direction. Green dots are AF22e measurements. All others are 43i measurements. All measurements here are 30-min averages.**


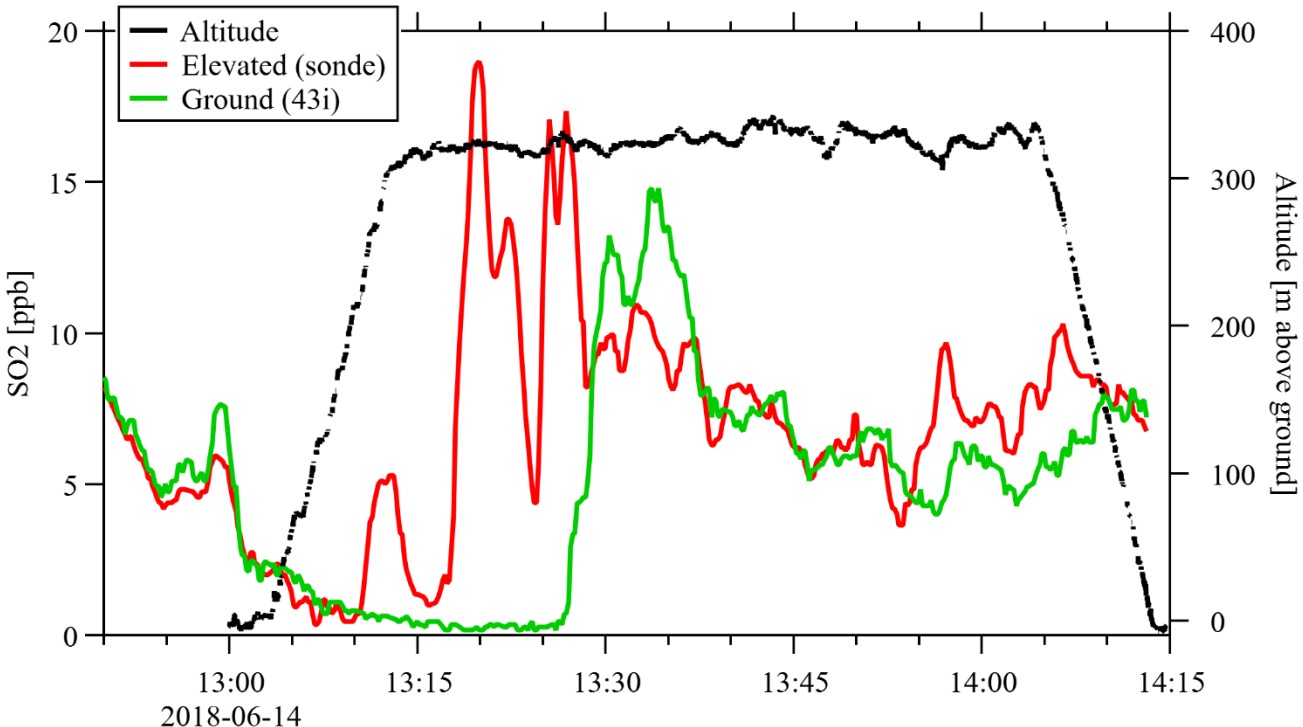

**Figure 6: SO₂ measurements on a tethered balloon flight (SO₂ sonde, red line) and ground level measurements (43i, green line). Balloon altitude (black line) also shown. Ground level is ~340 m ASL.**


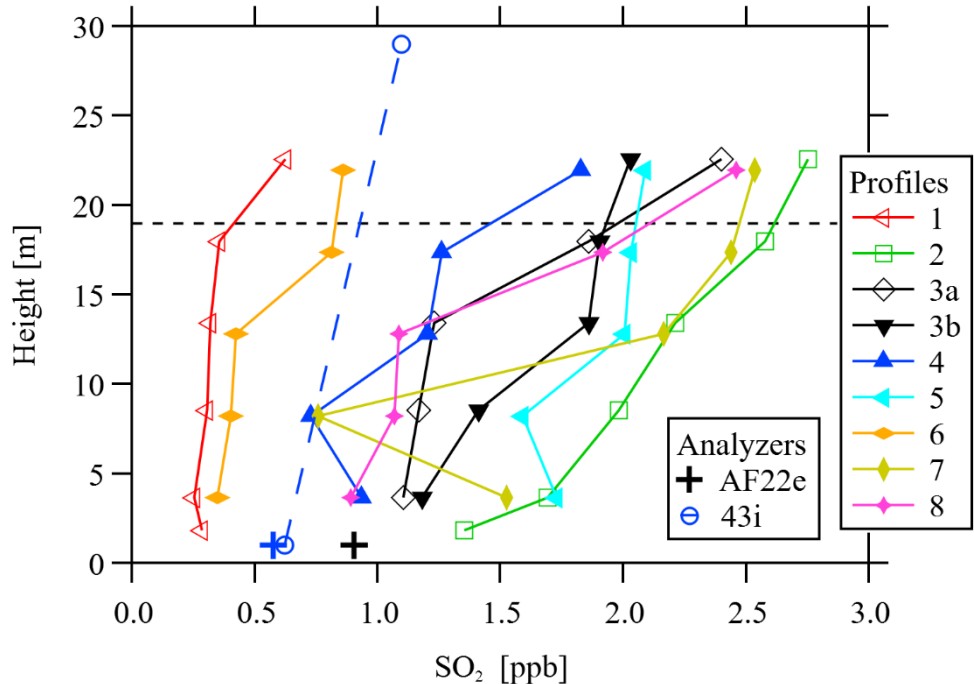

**Figure 7: Measurements of SO₂ mixing ratio with height. Profile numbers correspond to those listed in Table 1. Open symbols are profiles measured at YAJP. Closed symbols are profiles measured at site 1004. Circle (○) plot markers (blue with dashed line) show average measurements of two 43i instruments over a period coincident with Profile 4. Plus (+) plot markers show average measurement of AF22e instrument over two periods coincident with Profiles 3a and 3b (black) and Profile 4 (blue). The approximate canopy height is shown as a dashed horizontal line.**


**Table A1: WBEA sampling site information, listing station names, locations, and links to website description (last accessed Sept, 2022).**

| WBEA ID | Name | Lat. (N) | Lon. (W) | Website |
|---|---|---|---|---|
| AMS01 | Fort McKay Bertha Ganter | 57.1894 | 111.6406 | https://wbea.org/stations/bertha-ganter-fort-mckay/ |
| AMS06 | Patricia McInnes | 56.7514 | 111.4767 | https://wbea.org/stations/patricia-mcinnes/ |
| AMS07 | Athabasca Valley | 56.7334 | 111.3905 | https://wbea.org/stations/athabasca-valley/ |
| AMS17 | Wapasu | 57.2592 | 111.0386 | https://wbea.org/stations/wapasu/ |
| AMS18 | Stoney Mountain | 55.6214 | 111.1727 | https://wbea.org/stations/stony-mountain/ |

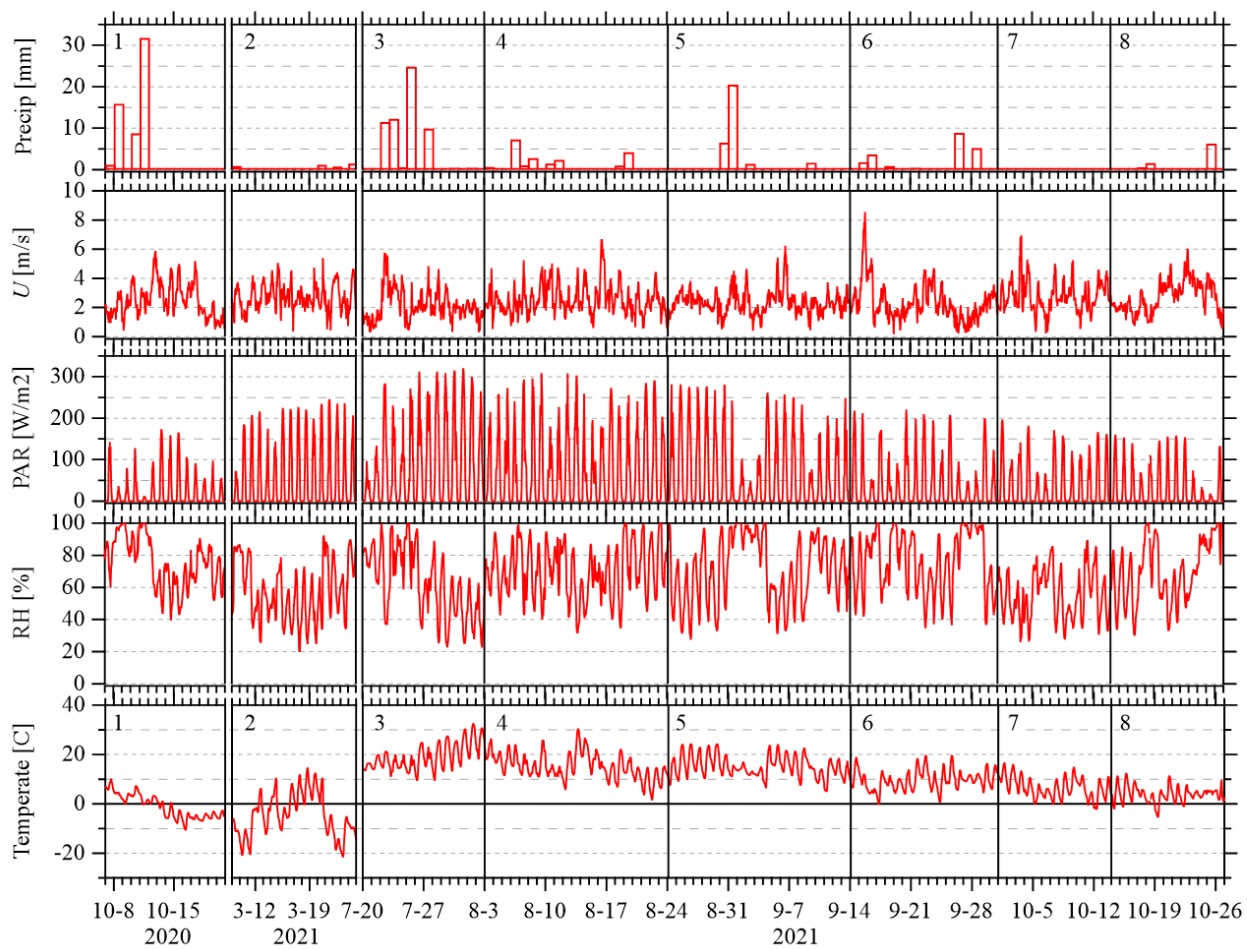

**Figure A1: Meteorological data during the 8 sampling periods. Daily precipitation is from Mildred Lake (ECCC), which is 12 km SW of the towers. Wind speed at 29-m (*U*), photosynthetically active radiation (PAR), relative humidity (RH), and temperature are from the 1004 tower.**