# Peer review of "High sulphur dioxide deposition velocities measured with the flux/gradient technique in a boreal forest in the Alberta oil sands region"

_Atmospheric Chemistry and Physics, 2022_

## Author Comment (AC1)

**Response to RC1: 'Comment on acp-2022-668', Anonymous Referee #1, 19 Dec 2022**

**General comments:**

This study presented a detailed estimation of SO2 dry deposition velocity in the Alberta oil sands region, and concluded that the possibility of much lower velocity in the model which determines the lifetime of SO2. Because our knowledge of the dry deposition velocity is still limited, I would like to generally agree to the publication of this manuscript. However, it is required to revise several points. Please see the following comments and address my concerns.

AC1.1: We thank the reviewer for providing this feedback.  Responses to specific points are listed below.

Many of the comments stem from confusion around the various corrections and adjustments to the deposition velocity, which were poorly explained in our original submission.  We have attempted to clarify by making use of different variables for different deposition velocities.  In the revised manuscript, $v_d$ refers to deposition velocity in a general sense, $v_{d,23m}$ refers to deposition velocity determined at a height of 23 m, $v_{d,40m}$ refers to deposition velocity adjusted to a height of 40 m (accounting for the aerodynamic resistance between 23 m and 40 m), and $v_{d,40m}^*$ refers to a corrected deposition velocity, accounting for a 30% overestimation due to the assumption of independent variables. We have added these variables throughout the manuscript and have taken care to differentiate between the terms "adjusted" (accounting for aerodynamic resistance between 23 m and 40 m) and "corrected" (accounting for a 30% overestimation due to the assumption of independent variables).

We have also noted that we used both "atmospheric resistance" and "aerodynamic resistance" to describe the same thing. In the revised manuscript, we have changed the text to only use "aerodynamic resistance" to avoid any confusion.

**Specific comments:**

- Line 16 (Abstract): It is better to explicitly express "corrected values" for these estimated velocities.

  AC1.2: We have added the qualifier "(when corrections are applied)" following the values.

- Line 19 (Abstract): In the main manuscript, it is stated that the study of Hayden et al. (2021) reported a value from 1.2 to 3.4 cm/sec. The value is different. Please confirm.

  AC1.3: The value 3.2 is a typo and should be 3.4.

- Line 31: Is this sentence indicates deposition "amounts" or "velocity"? Please clarify. In addition, I cannot get these values (1.7 and 5.4 times) from the values shown in Lines 32-33.

AC1.4: Added "velocities". The ratios of 1.7 and 5.4 were taken from Hayden et al. (19th line in Section 3.4). This appears to be a typo and 5.4 should be 5.9 (i.e. 3.4/0.58). We have modified the text to "5.9".

- Line 59 (Section 2.1): For readers, I feel the illustration of this location and its surrounding status will be helpful.

  AC1.5: We have added a new Figure 1 triptych with a Google map of the area and photos of YAJP and 1004 towers. Reference to this figure is added in Section 2.1.

- Line 183: Does the subscript for "r" generally stand for "aerodynamic" (www.atmos-chem-phys.org/acp/3/2067/)?

  AC1.6: We have modified this text to read "The total resistance to pollutant deposition at height $z$ ($r_{t,z}$) is modeled as the sum of the aerodynamic ($r_a$), quasi-laminar sublayer ($r_b$), and bulk surface ($r_c$) resistances. The deposition velocity is the inverse of the total resistance as … (Eq. 9)… where $C_z$ and $C_0$ are the concentrations at height $z$ and at a compensation point, respectively."

- Line 185: Duplicated equation number. Please correct.

  AC1.7: Equations 8 and 9 are corrected to numbers 9 and 10.

- Line 189: Because this is the key discussion, it will be better to have more explanations for GEM-MACH deposition parameterization. Especially, why this parameterization led to lower deposition velocities?

  AC1.8: We have added a new Subsection to discuss the GEM-MACH parameterization (Section 2.5) and move some of the text from Section 2.3 to this new section.

  "2.5 GEM-MACH Deposition Parameterization

  The GEM-MACH deposition parameterization used to compare to our measured values is described in Makar et al. (2018). The reader is referred to their Supplement S1 (their Equations S.1–S.20) for a detailed description. Very briefly, the parameterization accounts for aerodynamic ($r_a$), quasi-laminar ($r_b$), and bulk surface resistance ($r_c$), which includes resistances associated with soil, canopy, mesophyll, cuticle, and stomatal surfaces as well as resistance to buoyant convection. Here we model the forest as an evergreen needleleaf. The parameterization is a function of temperature, relative humidity, atmospheric stability, solar radiation, and $CO_2$ mixing ratio. For these values, we used measurements from the YAJP and 1004 towers and we calculated deposition velocity (herein $v_{d,\text{GEM}}$) for the time periods coincident with the profile measurements.

  As discussed above, the parameterized GEM-MACH deposition velocity values were significantly lower than the observations of Hayden et al (2021) for the same time periods and locations. Hayden et al. used a Monte Carlo analysis of the GEM-MACH deposition

algorithm to demonstrate that the most likely cause of underestimation was in the standard model assumption that concentration of hydrogen ions on the mesophyll, cuticle and exposed surfaces corresponded to a neutral pH (6.68). The Oil Sands facilities are known sources of significant base cation emissions (the neutralizing impact of the base cations on acidifying deposition was noted in Makar et al (2018)). Hayden et al (2021) showed that the increase in surface pH associated with deposited base cations could account for the discrepancy between modelled and measured $SO_2$ deposition velocities and fluxes. That is, $SO_2$ deposition close to the sources is likely being enhanced by the co-deposition of base cations."

We also add the following line to the last paragraph of the conclusions: "These results support the hypothesis discussed in Hayden et al. that $SO_2$ co-deposition with base cations may influence local $SO_2$ deposition fluxes."

- Line 241: The maximum value listed in Table 1 is "9.4". Please confirm.

  AC1.9: The value 9.3 is a type and has been corrected to "9.4".

- Line 261 (Section 4.1): This subsection has been well organized to seek uncertainties, but I have one question. When we see the list in Table 1, Profile 5 showed the distinctive lower deposition velocity. What caused this distinctive lower value?

  AC1.10: We have added the following text to Section 4.1 ($2^{nd}$ paragraph) "The deposition velocities ($v_{d,40m}$) from the 9 profiles can be compared to a normal distribution, with 78% (7 values) within one standard deviation ($\sigma$) of the mean, 1 value (Profile 8) $1.4\sigma$ from the mean and 1 value (Profile 5) $2.4\sigma$ from the mean. The anomalous deposition velocity of 2.9 cm s$^{-1}$ for Profile 5 is due to a combination of a weak gradient and high mixing ratio relative to the other profiles (Fig. 6). The fit to the profile is moderate ($R^2 = 0.72$), but not the weakest fit. Meteorological conditions shown in Appendix A (Fig. A1) demonstrate some rainfall ($> 25$ mm) and some cloudy, humid conditions, but similar conditions are seen in other profile periods (e.g. rain in profile periods 1 and 3, and clouds and high humidity in profile periods 6 and 8). Hence the reason for this anomalous value is unknown."

- Line 272: It will be better to explicitly state that these two values were mentioned as corrected velocities at 40 m.

  AC1.11: We have modified the text to "($v_{d,40m} = 6.9$ and 7.7 cm s$^{-1}$ for Profiles 1 and 8)" and have also changed "$v_d$" in 6 instances that follow in this section to "$v_{d,40m}$" to clarify that these values refer to the deposition velocities adjusted to 40 m.

- Line 347-349: I guess that this corrected range is presented based on the discussion in Section 4.1, but I cannot fully follow this correction. Please add final remarks in Section 4.1 to present which values were corrected from the discussion for a 30% overestimation.

  AC1.12: The following text is added to the end of Section 4.1: "Here we define a corrected deposition velocity (which assumes the 30% overestimation applies to all measurements) as

$v_{d,40\text{m}}^* = v_{d,40\text{m}} / 1.3$. Applying this correction to the range of deposition velocities listed in a Table 1 gives a corrected range of 2.1 to 5.9 cm s⁻¹, with an average of 4.6 cm s⁻¹."

**Technical corrections:**

Line 30 and elsewhere: Need subscript for "SO2".

AC1.13: This is corrected at line 30 and in the axis labels of Figs. 3, 4, 5, and 6.
* * *
**Response to RC2: 'Comment on acp-2022-668', Anonymous Referee #2, 05 Jan 2023**

The manuscript reports observations of $SO_2$ gradients in the region downwind of the Alberta Oil Sands, which the authors interpret to derive estimates of the dry deposition velocity. This is motivated by the inference of high rates of $SO_2$ deposition based on earlier aircraft flights in the region.

Overall, the data and analysis are interesting, but there are several assumptions used in the derivation of the flux and deposition velocities that are questionable. I would have found the manuscript easier to follow if some of these assumptions were highlighted in the introduction so that the purpose of some of the additional measurements (e.g. high time resolution data to assess the assumption of independent variables in the long time averages) was clear at the outset.

AC2.1: We thank the reviewer for their feedback and comments and for making suggestions which will make the manuscript easier to understand.

While the authors addressed the extent to which some of their assumptions introduce uncertainty into their calculations, I still missed an explanation of some aspects:

On lines 297-309, the authors use high frequency data from two heights to assess the assumption of independent variables. Does this exercise test the assumption that the concentration gradient and the momentum diffusion constant are independent values, or that the concentration and the deposition velocity are independent variables? Or both? For this subset of data, the authors find that violations of the assumption lead to a 30% overestimate in deposition velocity, but given the skewness of the data, can they be confident that this is a representative or conservative estimate?

AC2.2: Based on this feedback, we have renamed Subsection 2.3 to "Aerodynamic Resistance" and added Subsection 2.4, titled "Deposition Velocity Calculation". This moves the discussion of the time averaging and independent variables to its own section, which should make the analysis easier to follow. The first paragraph of this new section is as follows:

"The total deposition can be calculated combining Equations 8, 9, and 10. The use of long-term passive samplers (2 to 3 weeks in duration) to determine the gradients necessitates time-averaging the equations. If it is assumed that $K_M$ (a function of $u_*/\phi$), the concentration ($C$), and the gradient ($dC/dz$) are all independent variables, this gives

$$v_{d,40m} = \left(\left(\frac{\kappa z'_m}{Sc}\left\langle\frac{u_*}{\phi}\right\rangle\frac{1}{\langle C\rangle}\left\langle\frac{dC}{dz}\right\rangle\right)^{-1} + \frac{Sc}{\kappa}\left\langle\frac{\phi}{u_*}\right\rangle\ln\left(\frac{40}{23}\right)\right)^{-1}, \tag{11}$$

where the angle brackets $\langle\;\rangle$ indicate time-averaging over the sampling period. This assumes that there is no correlation between the stability-corrected friction velocity ($u_*/\phi$), the concentration ($C$), and the concentration gradient ($dC/dz$), since they are averaged separately in the equation. If $u_*/\phi$, $C$, and $dC/dz$ are correlated, the assumption of independent variables will introduce an error in this flux estimation (since $\langle u_*/\phi C\, dC/dz\rangle \neq \langle u_*/\phi\rangle\, 1/\langle C\rangle\, \langle dC/dz\rangle$). In order to estimate the error associated with the assumption of independent variables, we also calculate the deposition velocity (in Section 4.1) using a time series of 30-minute average, concurrent friction velocity, stability, and concentration measurements (using the high-frequency, SO$_2$ gradient measurements made with the two 43i instruments in August 2021), which does not require long-term averaging of these terms."

The start and end of the last paragraph of Section 4.1 (line 297-309 in the original submission) are also modified to refer to this new section and Equation 11 and to make the intention of the analysis clear. The modified paragraph begins and ends as…

"As discussed in Section 2.4, the assumption that $u_*/\phi$, $C$, and $dC/dz$ are independent variables will result in an error if they are correlated. We investigate this assumption by calculating the deposition velocity using a continuous time series of 30-min averages of the friction velocity, stability, and SO$_2$ observations (measured with the 43i instruments at 2 m and 29 m heights)." …
"Here we define a corrected deposition velocity (which assumes the 30% overestimation applies to all measurements) as $v_{d,40m}^* = v_{d,40m}\,/\,1.3$. Applying this correction to the range of deposition velocities listed in a Table 1 gives a corrected range of 2.1 to 5.9 cm s$^{-1}$, with an average of 4.6 cm s$^{-1}$."

Line 135 states that the flux gradient framework requires that the diffusion coefficient and the vertical gradient are constant throughout the canopy. But I believe is also an implicit assumption that the vertical flux is constant over the height of the measurements. Is that really true in a needleaf canopy where the elements leading to the uptake of the SO$_2$ are distributed over much of the vertical extent of the gradient? One can imagine significant differences in losses at different heights within the canopy that would manifest in the vertical structure differently depending on the timescales of turbulence/diffusion. Can these observations really be compared the deposition velocities obtained from a point high above the canopy in the constant flux layer, which would be most relevant to the aircraft data and the model parameterizations?

AC2.3: We agree that this implied assumption was not adequately addressed in the submitted manuscript. While we cannot estimate the uncertainty associated with this assumption within the scope of this work, we can point to two results that suggest the effect is not significant. Firstly, the profiles in Fig. 6 are relatively linear and do not show the curvature that would be associated with strong flux divergence. Secondly, the estimated deposition velocities show good agreement with the Hayden et al. (2021) flux/gradient measurements, which were made in the residential area of Fort McKay, more distance from immediate canopy effects. We have added text to discuss this in three sections as follows. In future we hope to investigate this effect using the onedimensional canopy model used in our companion paper (Zhang et al., 2022, ACPD, https://doi.org/10.5194/acp-2023-26)

Section 2.2 (after Eq. 3) "This also assumes that the flux divergence is insignificant in the canopy (equivalent to assuming all deposition is to the surface and not to the canopy elements). This assumption is discussed in Section 4.1."

Section 4.1 (5[th] paragraph) "The flux/gradient analysis outlined in Section 2.2 assumes that the vertical concentration profile is not significantly modified by deposition flux to canopy elements (such as pine needles). While the uncertainty due to this assumption is difficult to quantify, any strong flux divergence through the canopy should result in a consistent curvature of the concentration profile through the canopy. The profiles shown in Fig. 6 do not appear to demonstrate any consistent curvature away from a linear profile, suggesting that flux divergence may be minimal, and that deposition can be approximated following a "big-leaf" assumption."

Section 5 (4[th] paragraph) "The flux/gradient method does not account for flux divergence through the canopy and is equivalent to assuming that total deposition occurs near the surface only. Although the uncertainty associated with that assumption is difficult to quantify, we note that the vertical concentration profile shapes do not show any consistent curvature away from a linear gradient and that the deposition estimates show good agreement with the Hayden et al. tower measurements located in a relatively clear and residential area in Fort McKay. To quantify this uncertainty, the use of a high-resolution, one-dimensional canopy model (such as the model used in the Zhang et al., 2023) with varied deposition profiles is recommended for future studies."

The deposition velocity values derived by the authors are very high, implying minimal canopy resistance to deposition. In that case, we might expect the deposition velocity to reflect only atmospheric and quasi-laminar sublayer resistances. Can the authors confirm that such high deposition velocities are possible with estimates of those two constraints?

AC2.4: The aerodynamic and quasi-laminar sublayer resistances were included in a sensitivity analysis done by Hayden et al. (2021). The Hayden et al. analysis showed that the high values were in fact possible with existing theory and are likely associated with base cation deposition from the fugitive dust from the open pit mines. We have added an expanded description in a new section (2.5) to describe this.

"As discussed above, the parameterized GEM-MACH deposition velocity values were significantly lower than the observations of Hayden et al (2021) for the same time periods and locations. Hayden et al. used a Monte Carlo analysis of the GEM-MACH deposition algorithm to demonstrate that the most likely cause of underestimation was in the standard model assumption that concentration of hydrogen ions on the mesophyll, cuticle and exposed surfaces corresponded to a neutral pH (6.68). The Oil Sands facilities are known sources of significant base cation emissions (the neutralizing impact of the base cations on acidifying deposition was noted in Makar et al (2018)). Hayden et al (2021) showed that the increase in surface pH associated with deposited base cations could account for the discrepancy between modelled and measured $SO_2$

deposition velocities and fluxes. That is, $SO_2$ deposition close to the sources is likely being enhanced by the co-deposition of base cations."

Specific Comments

Lines 31-34 It would be useful to know how the aircraft data were interpreted to determine dry deposition velocities.

AC2.5: The following text is added: "Hayden et al. determined total deposition fluxes between multiple 2-dimensional (vertical and crosswind) flux screens created using interpolated aircraft-based wind and concentration measurements. The aircraft is flown in crosswind transects at various heights to determine the total advective flux passing through a screen, and the deposition flux is determined as the difference in advective flux between screens following a Lagrangian trajectory."

Figure 5 – The legend and the caption label the ground and sonde traces in opposite ways. Based on the text, the caption appears incorrect

AC2.6: We are very grateful to the reviewer for catching that oversight. The figure has been corrected.

---

## Author Response (AR2)

**Response to RC3: 'Comment on acp-2022-668', Anonymous Referee #3, 31 Mar 2023**

AC3.1: We wish to thank the reviewer for their feedback and for bringing the Meredith et al. (2016) study to our attention. We agree that we neglected to fully investigate and discuss the effect of the canopy on the use of the flux/gradient technique. We have modified the manuscript text including changes to Figure 1 and Table 1. We have added new information which we hope addresses these concerns and removed text with incorrect implications. Primarily, we have added deposition velocity calculation using the concentration gradient derived from the two highest (above-canopy) measurement heights. The average deposition velocity calculated with this 2-point, above-canopy gradient is higher than the deposition velocity calculated using all 5 measurement heights throughout and above the canopy, but it also shows much greater variation across the 9 profiles and the two averages are not statistically different (i.e. they are within < 1 standard error). Hence, we believe that our main conclusions are still supported by this modified analysis.

Herein any modified text within quotes is underlined (unless the entire quote is new). Line numbers refer to the newly revised manuscript.

**General comments:**

This study presents a detailed analysis of SO2 deposition velocity over a coniferous forest in Western Canada. This region is influenced by emissions from O&G and mining activities. The authors find that the removal rate is much faster than implied by a resistance-based model.

This study supports the findings of a recent study in the same region that used airplane and tower measurements (in an urban setting) to infer SO2 deposition rate.

AC3.2: We wish to clear up some confusion about the tower site from the Hayden et al. (2021) study, which is not an urban setting. We have added the text to the manuscript at line 42 as "Although the tower site was within the small town of Fort McKay (population 750), it was surrounded by wooded and grassy areas and can be considered a residential/rural site."

I have questions regarding some of the assumptions used to estimate the deposition velocity that need to be addressed before I can recommend publication.

**Major comments**

1. As pointed out by reviewer 2, the flux/gradient method generally requires that there be no source or sink within the region over which the gradient is estimated. As a result, the gradient is generally measured well above the canopy (see for instance Meredith, et al. (10.5194/amt-7-2787-2014) and references therein). In this study, the gradients are calculated using observations collected above and below the canopy (Fig. 2). This implications of this setup on the derivation of vd(SO2) need to be addressed very carefully given the body of literature that argues against it.

AC3.3: We have revised the paper to address this important issue. Text is added to Sections 2.2, 2.4, 3.3, 4.1, and the Conclusions. The LAI density profile is added to Figure 1 to demonstrate

the vertical distribution of the leaves at this site. A column is added to Table 1 in which the deposition velocity is calculated with the 2 above-canopy measurement heights. The figure below (not included in the revised manuscript) compares the LAI profile for our study with the LAI profile of the Harvard Forest where the Meredith et al. (2014) study was done. Although we define the canopy height in our study as 19 m based on the highest vertical extent of the trees, Meredith et al. (2014) define the canopy height based on the mean leaf-foliage value (giving a height of 18 m for that forest). Defining the canopy height in this way for the boreal forest in our study would give a height of ~11 m, demonstrating that our 2 highest measurements can also be considered well above the canopy.

[Figure]

Fig. 2 From Meredith et al., 2014 | Measurement heights and LAI profile at the YAJP and 1004 towers from this study

This figure (not included in the revised manuscript) compares the above-canopy measurement locations and LAI from Meredith et al. (2014) with the measurement heights and LAI from this study, demonstrating that the two highest measurement heights can be considered "above-canopy" relative to the forest leaf distribution.

Here we list a summary of the additional text:

Lines 146-157: "This approach has been demonstrated to reproduce deposition velocities by Wu et al. (2016) using gradients at heights of 16.5 m and 33 m in a 22-m high mixed-deciduous canopy. This mixed-deciduous forest had an LAI of 4.6, compared to the LAI of 1.17 at our boreal forest site, suggesting that the denser foliage would have a greater effect on the in-canopy gradient at the mixed-deciduous site relative to our boreal site. The approach was also demonstrated by Meredith et al. (2014) using gradients measured at heights of 24 m and 28 m in a nearly 24-m high temperate forest with an LAI of approximately 4. Here we determine the concentration gradient using a least-squares fit to the measured 5-point profile within and above the canopy. Although some profiles had 6 points, the lowest measurement is not used in these cases for consistency in the analysis. We also compare this to a 2-point concentration gradient determined using the two highest measurement heights. As the LAI density distribution in Figure

1d demonstrates, the two upper measurement heights (18 m and 23 m at YAJP or 17.5 m and 22 m at 1004) can be considered above-canopy relative to the canopy height of 19 m. Results from both gradient calculation techniques are compared in Section 3.3."

Lines 212-221: "As discussed in Section 2.2, the gradient ($dC/dz$) is determined using either a least-squares fit to 5 measurement heights (the 5-point gradient), or only using the two above-canopy measurement heights (the 2-point gradient). Using the 2-point gradient means that all uptake resistance ($r_{t,23m}$) is below the gradient. However, due to uncertainty in the measured value of $C$ using the passive samplers, there is higher uncertainty associated with a 2-point gradient measurement. This uncertainty can be reduced by calculating the gradient as a least-squares fit to the five values of $C$ at all measurement heights. However, there are likely sinks in the region over which the 5-point gradient is estimated. As Figure 1d demonstrates, most of the leaf area is closer to the surface and the mean canopy height (50% total LAI) is 11.5 m. Hence, the deposition velocity is calculated with the 5-point gradient assuming that the error in the calculated gradient due to sinks throughout the canopy is small compared to the uncertainty in a 2-point gradient measurement. Both approaches are compared in Section 3.3."

Lines 291-300: "The gradients ($dC/dz$) are determined either as a 2-point gradient above the canopy (to give $v_{d,23m}^+$) or from a least-squares fit to the 5-point gradient (to give $v_{d,23m}$). The mixing ratio ($C$) is determined from the highest sampler location. Using the 2-point, above-canopy gradients, the deposition velocities calculated with Eq. 8 range from 1.4 to 28.0 cm s$^{-1}$, with an average of 9.5 cm s$^{-1}$. Using the 5-point gradients, the deposition velocities calculated with Eq. 8 range from 2.9 to 9.4 cm s$^{-1}$, with an average of 6.9 cm s$^{-1}$. The $R^2$ values for the least-squares fits are given in Table 1. Although the above-canopy, 2-point gradient results in a higher average deposition velocity, there is much higher variation across the 9 profile measurements, and the average values from each method (9.5 cm s$^{-1}$ and 6.9 cm s$^{-1}$) are not statistically different at a 55% confidence level (i.e. $\overline{v_d} \pm 0.75\ \sigma/\sqrt{n}$). Hence, we conservatively focus on the lower deposition velocities calculated with the 5-point gradient determined by least-squares fit ($v_{d,23m}$), but note the high uncertainty associated with these measurements."

Lines 331-337: "As discussed above, the use of the 5-point gradient effectively moves the total resistance (including aerodynamic, quasi-laminar sublayer, and bulk surface resistances) to the ground level, following the "big-leaf" assumption typically used by regional and global-scale models, as opposed to a vertical distribution of uptake throughout the canopy. While there is uncertainty associated with this assumption which is difficult to quantify, the average deposition velocity calculated with the 2-point, above-canopy gradient is greater than the average deposition velocity calculated with the 5-point gradient, suggesting that the deposition velocity calculated with the 5-point gradient may be a conservative estimate (although the averages are within $\pm 0.75$ standard errors and hence not significantly different)."

Lines 449-459: "The flux/gradient method does not account for flux divergence through the canopy and is equivalent to assuming that the total deposition occurs near the ground level only. Although the uncertainty associated with that assumption is difficult to quantify, calculating the gradient using above-canopy measurements results in a higher average deposition velocity and a greater disparity between our measurements and those determined by Hayden et al. (2021). However, using the above-canopy measurements, there is high variability between the deposition

velocities calculated from the different profiles, suggesting high uncertainty in the average measured value. We note that the deposition velocities calculated with the 5-point measurement (throughout and above the canopy) show good agreement with the Hayden et al. (2021) tower measurements located in a relatively clear and residential/rural area in Fort McKay, suggesting that the error associated with the vertical positioning of the uptake elements may be small. To quantify this uncertainty, the use of a high-resolution, one-dimensional canopy model (such as the model used in the Zhang et al., 2023) which can correctly model the vertical distribution of uptake is recommended for future studies."

The authors suggest that the lack of curvature in the SO2 profile implies that SO2 removal by needles is minimal. This would be quite surprising.

AC3.4: This text was confusing and has been removed. There is no longer any reference to the lack of curvature in the profiles.

This could also mean that some of the assumption used are not correct. For instance, I am surprised that du/dz is assumed to be constant throughout the canopy (see for instance, https://rmets.onlinelibrary.wiley.com/doi/epdf/10.1002/qj.49709741414).

AC3.5: We have changed the wording here (at line 161) as "…the wind speed gradient ($du/dz'$) is approximated as $\Delta u/\Delta z$ from the wind velocity difference between heights of 29 m and 5.5 m. The uncertainty due to this approximation is investigated below." This is followed (at line 172) by "A least-squares fit to all the 30-min values of $K_{M,P}$ as a function of $K_{M,G}$ over the ~10-month period gave a slope of 2.6 with $R^2 = 0.83$ (Fig. 3), which supports the use of the $\Delta u/\Delta z$ approximation."

Although the wind profile within a canopy is unlikely to be linear, this direct comparison of the diffusion coefficients derived using either $u_*(K_{M,P})$ or the linear gradient ($K_{M,G}$) demonstrates that the two values compare well (See Fig. 3).

2. The authors assumptions for flux/gradient require that R_needle >>1

If I am not mistaken, the GEM-MACH model would imply that the maximum deposition velocity = 1/Rcan (Table S3, Makar (2018)) which for evergreen yields vd_max ~1 cm/s or 2-5x slower than observed.

This would suggest that significant removal of SO2 needs to happen within the canopy to explain the observed vd(SO2). Such removal would contradict the assumption made to estimate vd(SO2).

This discrepancy needs to be addressed.

AC3.6: As stated in our responses above, we did not mean to imply that the removal by needles in minimal. As quoted above (line 332) "…the use of the 5-point gradient effectively moves the total resistance (including aerodynamic, quasi-laminar sublayer, and bulk surface resistances) to

the ground level, following the "big-leaf" assumption typically used by regional and global-scale models, as opposed to a vertical distribution of uptake throughout the canopy."

Now that the deposition velocity is verified using the above-canopy, 2-point gradient, the assumption of R_needle >> 1 is not required and there is no contradiction.

**Minor comments**

Line 12. Human exposure to what? (SO4?)

AC3.7: For clarity, we have removed "and human exposure" from this sentence.

Line 15. "predict". I would suggest to rephrase to clarify that vd(SO2) is not directly observed

AC3.8: We change "predict" to "infer".

Line 19. The statement that the estimated vd(SO2) is close to aircraft observations is a bit confusing. The numbers quoted on line 16 are ~2x greater than the aircraft estimates (line 19). This also holds for line 274.

AC3.9: We agree this is a better way to present the comparison. We were thinking of it as two ranges (1.2-3.4 and 2.7-7.7) which overlap in the 2.7-3.4 range (and are hence "near" or "similar"). As noted by the reviewer, the minimum (and maximum) of each range differs by a factor of 2 so that is indeed a better comparison method.

We have modified the text in the Abstract as "Accounting for these uncertainties, the range of measurements is approximately double the previous aircraft-based measurements (1.2–3.4 cm s⁻¹) and are more than 10 times higher than model estimates for the same measurement periods (0.1–0.6 cm s⁻¹)."

The text at line 301 (formerly line 274) is modified as "The values measured here are approximately double those of Hayden et al…".

Line 21. Please specify that this conclusion only applies to the type of environments investigated here.

AC3.10: The text at line 20 is modified to "…suggesting that $SO_2$ in the AOSR region has a much shorter lifetime in the atmosphere than is currently predicted by models." We also modified the conclusions (at line 460) from "… to the environment" to "… to the type of environment studied here".

AC3.11: As a further note (not a response to a specific reviewer comment), because of the added discussion comparing the "2-point" above-canopy gradient to the 5-point gradient, we have modified the description of the $SO_2$ gradient measurements made by gas analyzers above and below the canopy (at 2 heights). While these were referred to as "2-point" measurement in the previous version (primarily in the Conclusions), we now refer to them as "continuous"

measurements to distinguish them from the passive samples while avoiding confusion with the 2-point, above-canopy passive sampler measurements.

---

## Author Response (AR3)

As per the editor request, the figure from the response to reviewers has been included in the submission as supplementary information.